# Compensating transport trends in the Drake Passage frontal regions yield no acceleration in net transport

Manuel O. Gutierrez-Villanueva[1] ✉, Teresa K. Chereskin [1] & Janet Sprintall [1]

Although the westerly winds that drive the Antarctic Circumpolar Current (ACC) have increased over the past several decades, the ACC response remains an open question. Here we use a 15-year time series of concurrent upper-ocean temperature, salinity, and ocean velocity with high spatial resolution across Drake Passage to analyze whether the net Drake Passage transport has accelerated in the last 15 years. We find that, although the net Drake Passage transport relative to 760 m shows insignificant acceleration, the net transport trend comprises compensating trends across the ACC frontal regions. Our results show an increase in the mesoscale eddy activity between the fronts consistent with buoyancy changes in the fronts and with an eddy saturation state. Furthermore, the increased eddy activity may play a role in redistributing momentum across the ACC frontal regions. The increase in eddy activity is expected to intensify the eddy-driven upwelling of deep warm waters around Antarctica, which has significant implications for ice-melting, sea level rise, and global climate.

The Antarctic Circumpolar Current (ACC) is one of the critical components of the Southern Ocean (SO) and the global ocean climate system. The ACC flows eastward unbounded around Antarctica, connecting the three major ocean basins, promoting exchange between them, and allowing the establishment of a global-scale overturning circulation[1]. The ACC and the overturning circulation transport heat, salt, carbon dioxide, and other tracers around the globe and strongly influence the Earth's climate. Therefore, studying the three-dimensional structure of the ACC and its properties is pivotal to assessing how the SO is responding to climate change.

An intensification of the SO westerly winds that drive the ACC has been documented over the past several decades, reflecting an intensification of the Southern Annular Mode (SAM; see refs. 2,3), the leading mode of atmospheric variability in the SO. Modeling studies show that the positive trend in the winds is a response to both depletion of the atmospheric ozone and increments in greenhouse gas emissions over the last decades[4–6]. The intensification in eastward wind stress, and therefore northward Ekman transport[1], increases the horizontal density gradients that characterize the fronts that comprise the ACC. In coarse-resolution global climate models, the steepened

gradients accelerate the flow in the ACC fronts, resulting in increased model ACC transport[7,8]. In contrast, eddy-permitting global climate models show that the enhanced available potential energy resulting from the steepened isopycnals is released through baroclinic instabilities, which feed into the vigorous eddy field within the ACC, reaching an eddy saturation state, i.e., eddy activity increases as the wind stress increases with no net acceleration of the ACC[9–15].

Overall the SO is a meagerly sampled section of the global oceans as it represents a challenge to observe due to inhospitable conditions and remoteness. An exception is Drake Passage, which has historically provided an ideal location for monitoring the ACC transport[16–20], as it represents the narrowest constriction (800 km) through which the ACC navigates on its circumpolar path (Fig. 1a). As such, Drake Passage transport and its long-term variability serve as valuable metrics for validating ocean and global climate models. However, to date, none of the observational studies have found a clear trend in the net Drake Passage transport[10,17–20]. Moreover, these studies were unable to explore trends within the ACC frontal regions owing to the lack of direct current velocity measurements at a sufficient spatial resolution to resolve the frontal regions and mesoscale activity. The lack of

[1]Scripps Institution of Oceanography, University of California San Diego, 9500 Gilman Dr, La Jolla, CA 92093, USA. ✉e-mail: mog008@ucsd.edu

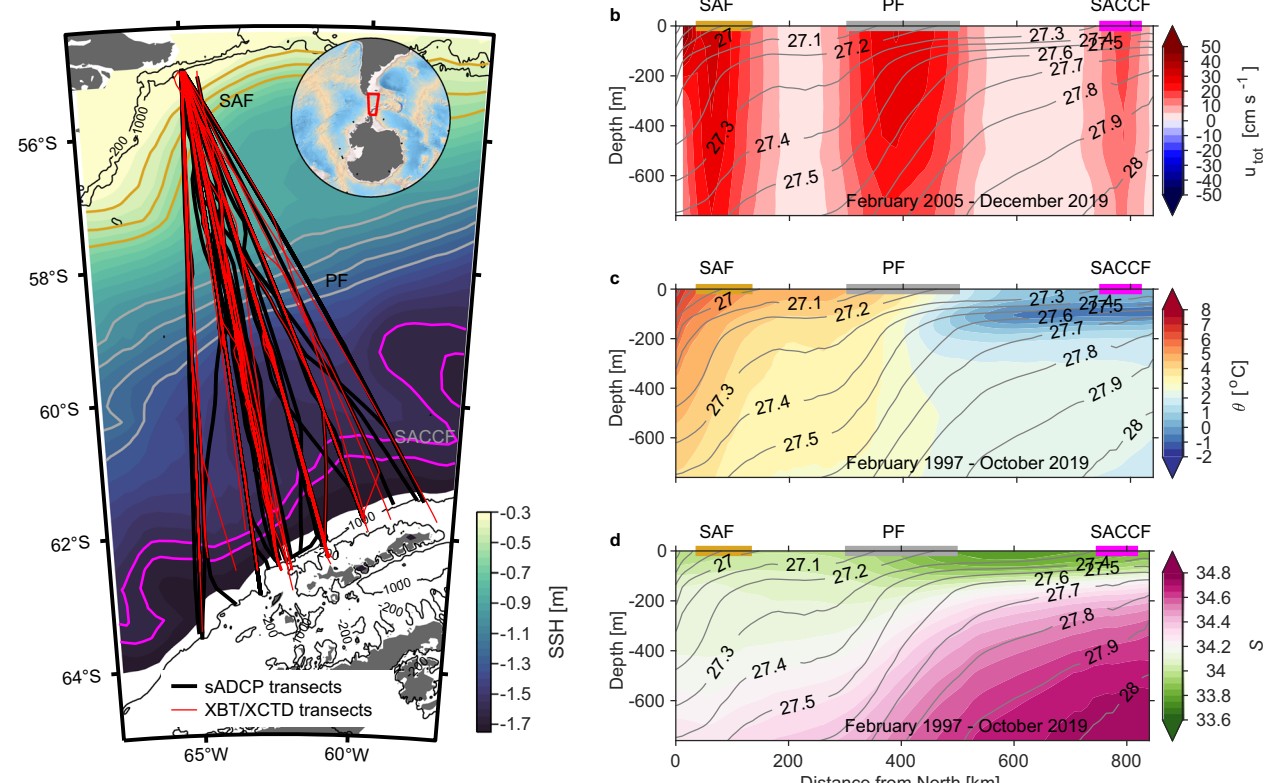

**Fig. 1 | Drake Passage transects and mean cross-transect velocity, potential temperature, and salinity. a** The Drake Passage observations between the South American and Antarctic shelf breaks. Shipboard Acoustic Doppler Current Profiler (sADCP) velocity transects from February 2005 to April 2019 and expendable bathythermograph/expendable conductivity–temperature–depth (XBT/XCTD; temperature/salinity) transects from February 1997 to October 2019 are shown in black and red lines, respectively. Filled contours are the mean sea surface height (SSH) [m] from ref. 50 with a 0.60 m constant subtracted. The mean sea surface drops by 1.65 m from South America to Antarctica across Drake Passage. Gold, gray, and magenta contours show the mean locations of the Subantarctic Front (SAF), Polar Front (PF), and Southern Antarctic Circumpolar Current Front (SACCF) respectively[28]. The 200-m and 1000-m isobaths are included in solid black contours. Climatological mean **b** cross-transect velocity $u_{tot}$ [cm s$^{-1}$], **c** potential temperature $\theta$ [°C], and **d** salinity $S$ across Drake Passage. All (114) temperature/salinity and (237) velocity transects are used to estimate the record-length mean. Mean neutral density surfaces (kg m$^{-3}$)[49] are shown in gray contours. Distance increases moving towards the Antarctic shelf. Horizontal gold, gray, and magenta bars indicate the mean locations of the SAF, PF, and SACCF, respectively.

long-term in situ observations (velocity, temperature, salinity) with high horizontal resolution in the SO hampers the exploration of trends in the ACC transport and properties across the different frontal regions.

This study uses a unique observational time series of year-round near-repeat upper-ocean temperature, salinity, and velocity across the Drake Passage (Fig. 1a) to explore whether the Drake Passage transport in the upper 760 m shows significant acceleration in the last 15 years. The repeated upper-ocean velocity transects provide the most efficient way to measure the upper-ocean transport[21]. The high along-track spatial resolution of the transects allows us to observe patterns on the order of the first baroclinic Rossby radius (20–10 km)[22]. Surprisingly we found that, although the net Drake Passage transport relative to 760 m shows insignificant acceleration, there are compensating trends across the ACC frontal regions. Our results show an increase in the mesoscale eddy activity between the fronts due to buoyancy changes in the fronts consistent with an eddy saturation state.

## Results

### Climatology of cross-passage velocity, temperature, and salinity
Climatological mean sections of cross-transect velocity $u_{tot}$, potential temperature $\theta$ (temperature hereinafter), and salinity $S$ across Drake Passage using all available velocity (237) and temperature/salinity (114) transects (see Methods) are shown in Fig. 1b–d. The cross-transect velocity climatology reveals the jets associated with the main ACC

fronts (Fig. 1b). The narrow Subantarctic Front (70–150 km, Fig. 1b) is the strongest jet with mean velocities reaching 60 cm s$^{-1}$ near the surface (Fig. 1b). Located south of the Subantarctic Front, the jet associated with the Polar Front extends for almost 200 km across Drake Passage and has a weaker mean velocity than the Subantarctic Front (300–500 km; Fig. 1b). Further south (750–800 km) the weakest ($u_{tot} < 15$ cm s$^{-1}$) of the three jets is the Southern ACC Front (Fig. 1b).

Relatively warm water masses ($\theta > 2$ °C) are located northward of the Polar Front (<400 km; Fig. 1c). The Subantarctic Front marks the region with the strongest temperature stratification and lateral gradients (Fig. 1c). The northernmost subsurface tongue of cold, fresh Antarctic Winter Water (AWW; $\theta < 1$ °C, $S \sim 34$) marks the location of the Polar Front, the boundary with the warm subtropical waters (Fig. 1c, d). Below the AWW (<−200 m), the dense Upper Circumpolar Deep Water (UCDW; $\theta \sim 2$ °C, $S > 34.40$) is very homogeneous in temperature (Fig. 1c, d).

### Trends in Drake Passage transport
To explore long-term trends, we estimated the time series of Drake Passage transport in the upper 760 m. We defined Drake Passage total transport $U_{tot}$ as the integral of the cross-track velocity $u_{tot}$ from 0 to 760 m depth and from 0 to ~850 km across Drake Passage. When temperature and salinity observations were available, the geostrophic velocity $u_{geo}$ and therefore the geostrophic transport (density-driven, depth-dependent relative to 760 m; $U_{geo}$) could also be estimated. For the subset of 60 coincident velocity and temperature/salinity transects

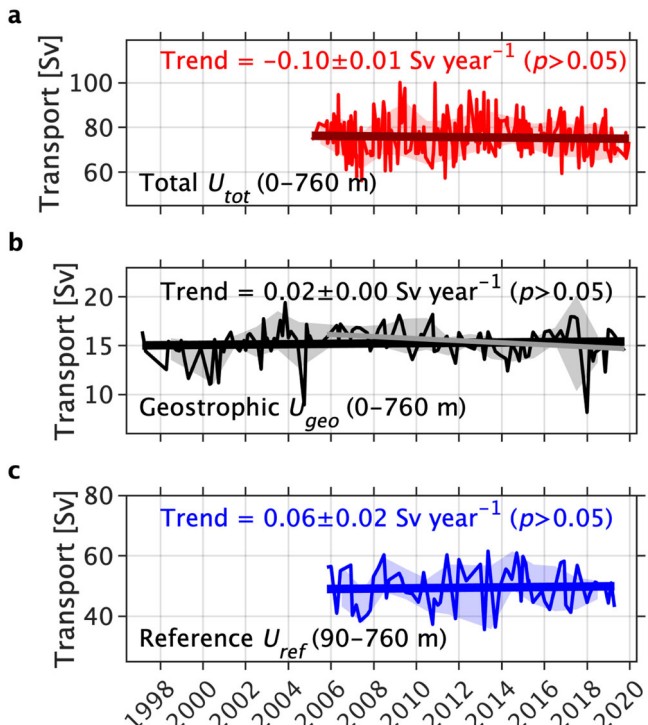

**Fig. 2 | Time series and trends of the Drake Passage transport [Sv] in the upper 760 m.** Time series of net cross-transect **a** total $U_{tot}$, **b** geostrophic $U_{geo}$, and **c** reference $U_{ref}$ Drake Passage transport ($1\,Sv = 10^6\,m^3\,s^{-1}$). The seasonal cycle (annual + semiannual harmonics) was removed for each time series. Standard deviations for each calendar year are shown by shading. Thick solid lines show the linear fits for the time series. Gray solid line in (**b**) is the linear fit for the period of October 2005–April 2019, coinciding with (**c**) reference transport period. Least-squares trends and 95% error bars are shown with text [Sv year⁻¹]. None of the trends are statistically significant after the modified Mann–Kendall test ($p > 0.05$).

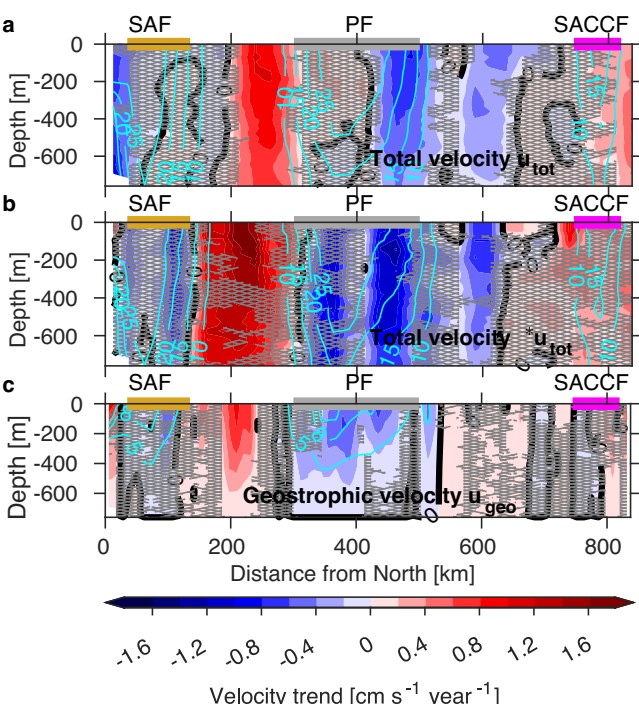

**Fig. 3 | Trends in cross-transect velocity components [cm s⁻¹ year⁻¹] from October 2005–April 2019.** Trends in cross-transect **a**, **b** total $u_{tot}$, and **c** geostrophic $u_{geo}$ component. Trends in (**a**) are estimated using all available (237) velocity transects while trends in (**b**) are calculated using only those (60) velocity transects that have a concurrent temperature/salinity transect used to compute the geostrophic $u_{geo}$ component in (**c**). Black thick solid contours show the zero trend. Hatched areas are nonsignificant trends at 95% confidence after the modified Mann–Kendall test ($p > 0.05$). Cyan contours show the mean velocity component [cm s⁻¹]. The mean location of the Subantarctic Front (SAF), Polar Front (PF), and Southern Antarctic Circumpolar Current Front (SACCF) are indicated by horizontal gold, gray, and magenta bars, respectively.

occupied from October 2005 to April 2019, the total Drake Passage transport based on direct velocity observations was decomposed into a geostrophic component plus a residual ($U_{ref}$, depth-independent; see "Methods"). Hereinafter, we interpret the acceleration and deceleration of the eastward transport and flow as positive and negative trends, respectively. Trends are statistically significant when the probability $p < 0.05$ after the modified Mann–Kendall test (see "Methods").

None of the Drake Passage transport time series show statistically significant acceleration over the last two decades ($p > 0.05$; Fig. 2). Trends were statistically insignificant regardless of the period used or number of transects used (Supplementary Fig. S1). Our results are consistent with previous Drake Passage observational studies using coarser resolution in situ observations[17,18] and altimetry-derived geostrophic currents[20] which found no significant acceleration of the Drake Passage transport. In the following subsection, we explore an alternative hypothesis to explain the lack of trends in the net Drake Passage transport.

**Trends in transport across Drake Passage**

We investigated whether there are opposing regional trends in the ACC frontal jets consistent with no trend in Drake Passage transport. We estimated trends in the cross-transect total $u_{tot}$ and geostrophic $u_{geo}$ velocity components for the common sampling period October 2005–April 2019.

Significant opposing trends in cross-track velocity are found within Drake Passage. All significant trends ($p < 0.05$) show no change in sign with depth (Fig. 3). Since the jets of the ACC are equivalent barotropic (velocity is self-similar with depth; see ref. 23), the sign of the trends likely holds over the entire water

column, although the magnitude may decay. The largest significant acceleration of the eastward flow is found in the region between the Subantarctic Front and Polar Front for the total and geostrophic components (150–300 km, Fig. 3a, c). A weak, significant acceleration is also found in the Southern ACC Front (>750 km, Fig. 3a, c). In contrast, a significant deceleration of the total flow component is found within the Polar Front (400–500 km, Fig. 3a) and to the south (550–600 km, Fig. 3a). A broader and weaker deceleration is found in the geostrophic component, but mainly within the Polar Front (350–500 km, Fig. 3c). A similar distribution in space and sign is found (although with slightly larger amplitudes) when total velocity trends are calculated with the subset of the total velocity transects that coincide with the temperature/salinity transects (Fig. 3b).

We estimated the time series of cross-transect total $u_{tot}$, geostrophic $u_{geo}$, and reference $u_{ref}$ transport per distance bin. The net total, geostrophic, and reference Drake Passage transports are $75.54 \pm 0.64$ Sv, $15.85 \pm 0.20$ Sv, and $51.08 \pm 0.85$ Sv, respectively (Fig. 4b, e, h). The sum of the geostrophic, reference and Ekman ($8.09 \pm 0.22$ Sv; see "Methods") transports yields $75.02 \pm 0.90$ Sv, which agrees with the net total transport calculated using the concurrent transects ($77.87 \pm 1.04$ Sv; green line in Fig. 4b). Extending the total transport calculation to the upper 970 m yields a 15-year mean of $89.41 \pm 4.69$ Sv, in close agreement with ref. 24's mean of $95.00 \pm 2.10$ Sv based on the first five years of data (only using transects along the most repeated line), and the 40-year mean modeled transport of 92.7 Sv in the upper 1000 m calculated in ref. 21 from a data-constrained eddy-permitting ocean model. The 75.54 Sv net total

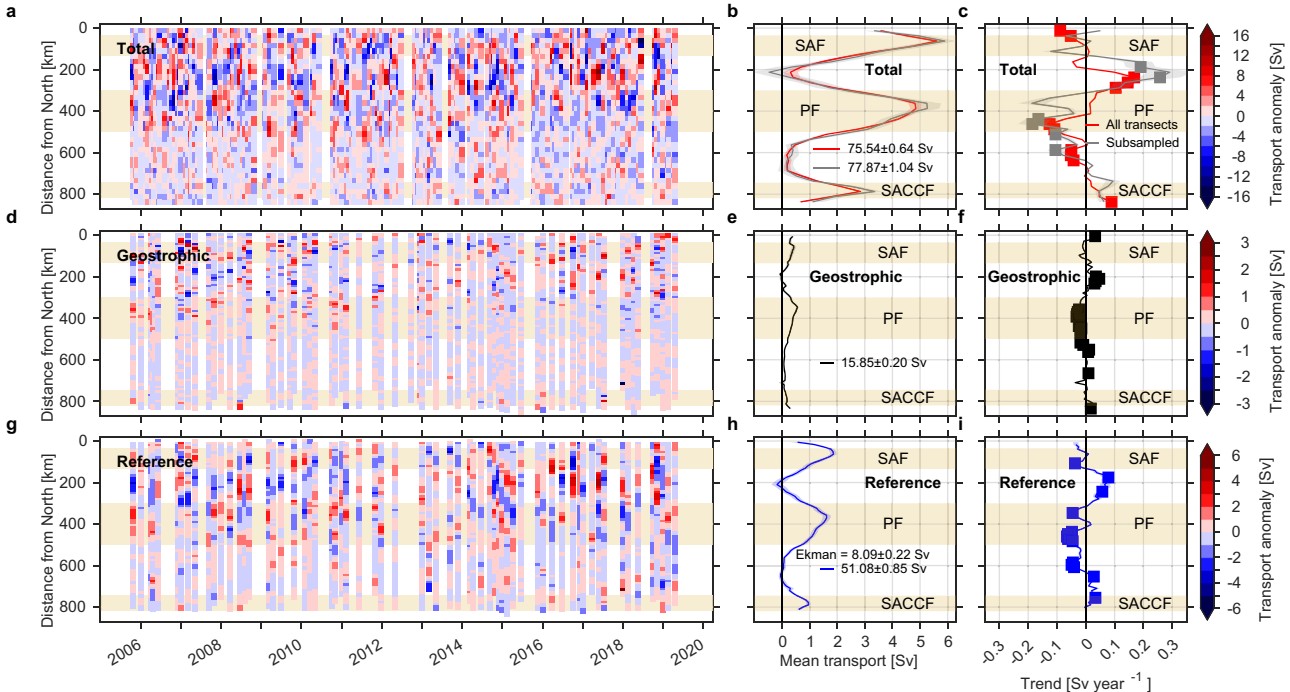

**Fig. 4 | Time series of transport anomaly [Sv], mean transport [Sv], and trends [Sv year⁻¹] per distance bin [km] for October 2005–April 2019.** Time series of **a** total, **d** geostrophic, and **g** reference transport anomalies in the upper 760 m. Mean **b** total, **e** geostrophic, and **h** reference transport [Sv] per distance bin. Trends in **c** total, **f** geostrophic, and **i** reference transport [Sv year⁻¹] per distance bin. Red and gray lines (**b**, **c**) show the mean and trends estimated for all (237) available transects and only those (60) velocity transects that have a coincident temperature/salinity transect, respectively. Filled squares (**c**, **f**, **i**) indicate that trends are significant at 95% confidence after the modified Mann–Kendall test ($p < 0.05$). Total

transport anomalies (**a**) are estimated based on the mean using all available transects (red line). Standard errors for each line are shown in shaded areas in (**b**, **c**, **e**, **f**, **h**, **i**). Mean ( ± standard error) total, geostrophic and reference transport integrated across Drake Passage are indicated with text in (**b**, **e**, **h**). Mean cross-track Ekman transport (0–90 m) is also indicated in (**h**) (see "Methods"). Shaded rectangles show the approximate distance intervals for the Subantarctic Front (SAF), Polar Front (PF), and Southern Antarctic Circumpolar Current Front (SACCF)[28].

transport in the upper 760 m is 56% of the full-depth total canonical transport of 134 Sv estimated in ref. [17] and 48% of the 40-year mean full-depth model transport of 157 Sv estimated in ref. [21].

Transport anomalies for the three components are the largest in the northern half of Drake Passage, especially in the region between the Subantarctic Front and Polar Front (0–500 km, Fig. 4a, d, g). This region has the largest mesoscale eddy activity (meandering of the fronts and eddy formation) within Drake Passage[25–28]. South of this region the transport anomalies are the weakest (>500 km, Fig. 4a, d, g). While the time series of transport anomalies do not readily show discernible trends, the estimated trends (Fig. 4c, f, i) show clear significant trends consistent with the velocity trends (Fig. 3). All transport components show a clear acceleration of the region between the Subantarctic Front and Polar Front (150–300 km; Fig. 4c, f, i) and deceleration of the southern flank of the Polar Front (350–550 km; Fig. 4c, f, i). Similar patterns are found when total transport trends are calculated with the total velocity transects that coincide with the temperature/salinity transects (Fig. 4c), or using the transects falling along the most repeated crossing (not shown). The geostrophic and reference transport trends (Fig. 4f, i) summed across Drake Passage yield − 0.10 Sv year⁻¹ and 0.04 Sv year⁻¹, respectively, and are in close agreement with the insignificant trends calculated from the time series of net Drake Passage transport (Supplementary Fig. S1). The sum of the total transport trends (Fig. 4c) yields 0.08 Sv year⁻¹, which is of the opposite sign (but insignificant) compared to that estimated from the net transport time series (Supplementary Fig. S1).

The results based on binning the transport by distance indicate that the different ACC frontal regions in Drake Passage have tended to compensate each other over the last 15 years such that the net Drake Passage transport shows no significant acceleration.

Nonetheless, the acceleration of the region between the fronts could be an effect of an increase in the meandering of the ACC frontal jets and eddy formation, an aspect not captured when using the fixed cross-passage coordinate system. To explore this hypothesis we constructed time series of transport anomalies and trends analogous to Fig. 4, but where the binning was done per pair of streamlines (Fig. 5), using synoptic sea surface height (SSH) derived from a combination of daily altimetric sea-level anomalies and mean geostrophic streamlines (Fig. 1a; see "Methods"). Contours of SSH provide a natural coordinate system (streamlines) for geostrophic flows. The streamlines allowed us to track the position of the fronts that comprise the ACC, which can vary (both in position and distance between each pair of streamlines) daily due to strong meandering of the fronts and eddy formation[25, 28].

As with the time series of transport anomalies in passage coordinates (Fig. 4a, d, g), the transport anomalies in streamwise coordinates (Fig. 5a, d, g) are largest in northern Drake Passage, but are concentrated north of the Polar Front. Similar to the time series in passage coordinates (Fig. 4a, d, g), trends in the streamwise time series are difficult to discern (Fig. 5a, d, g). The magnitude and pattern of the estimated trends for the total transport in streamwise coordinates (Fig. 5c) are much reduced relative to passage coordinates (Fig. 4c), suggesting that at least some of the apparent acceleration in the zone between the Polar and Subantarctic Fronts (Figs. 3 and 4c) was due to fronts meandering and eddies. When binned by streamline, the transport anomalies within meanders and eddies contribute to the anomalies of the fronts rather than the interfrontal zone. The main significant trend in the streamwise coordinate system is that the Subantarctic Front has significantly accelerated over the last 15 years (Fig. 5c). This result is only evident when using the streamwise

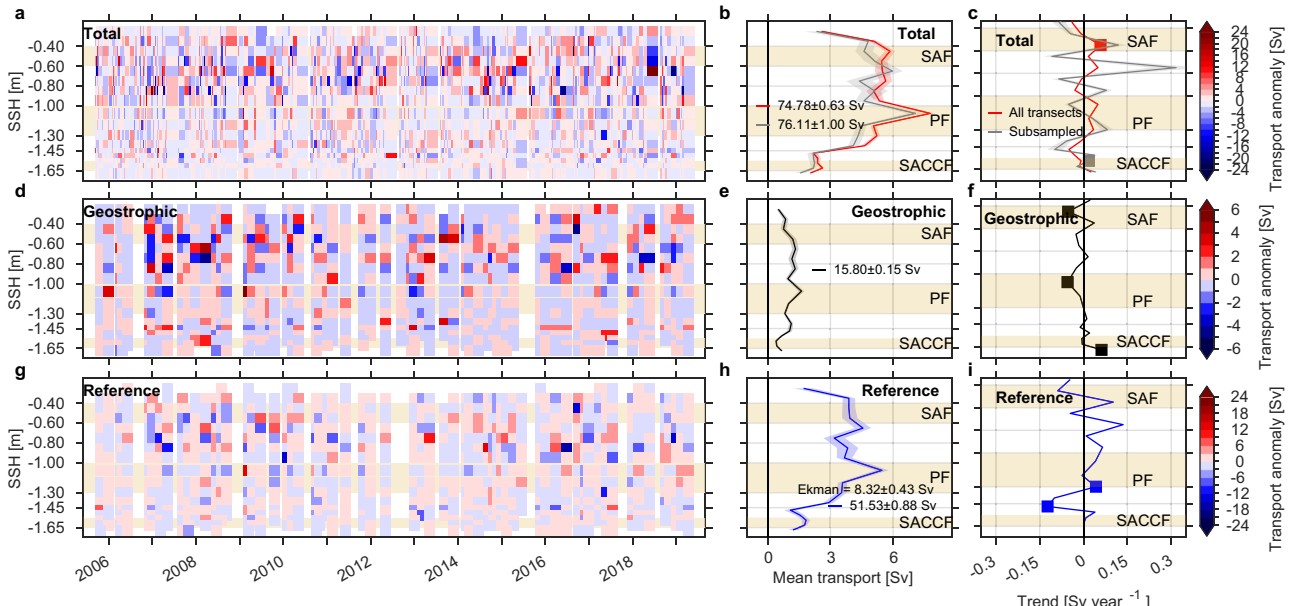

**Fig. 5 | Time series of transport anomaly and mean transport [Sv] per pair of sea surface height (SSH) streamlines [m] for October 2005–April 2019.** Time series of **a** total, **d** geostrophic, and **g** reference transport anomalies in the upper 760 m. Mean **b** total, **e** geostrophic, and **h** reference transport [Sv]. Trends in **c** total, **f** geostrophic, and **i** reference transport [Sv year⁻¹]. Red and gray lines (**b, c**) show the mean and trends estimated for all (237) available transects and only those (60) velocity transects that have a concurrent temperature/salinity transect, respectively. Filled squares (**c, f, i**) indicate that trends are significant at 95% confidence after the modified Mann–Kendall test ($p < 0.05$). Total transport anomalies (**a**) are estimated based on the mean using all available transects (red line). Standard errors for each line are shown in shaded areas in (**b, c, e, f, h, i**). Mean ( ± standard error) total, geostrophic and reference transport integrated across Drake Passage are indicated with text in (**b, e, h**). Ekman transport is indicated in (**h**) (see "Methods"). Shaded rectangles show the streamwise intervals for the Subantarctic Front (SAF), Polar Front (PF), and Southern Antarctic Circumpolar Current Front (SACCF)[28].

coordinate system, which is able to capture the effect of the steering of the fronts due to meandering and eddy detachment, an aspect that is missed when using the passage coordinates (Fig. 4).

We will explore the increasing mesoscale activity hypothesis in "Trends in eddy activity" by estimating trends in eddy kinetic energy and momentum flux. In the following subsection, we explore buoyancy changes across Drake Passage that may contribute to increased eddy formation.

**Buoyancy changes across Drake Passage**

To understand buoyancy changes across the ACC frontal regions in Drake Passage, we estimated trends in the temperature and salinity in the upper 760 m and decomposed them into along neutral density layers (spice), vertical movement of density layers (heave), and total (depth) trends[29, 30] (see "Methods").

Significant warming and salinification due to spice is found in the Subantarctic Front (< 200 km; Fig. 6a, d). Increased heat uptake from the atmosphere to the ocean due to increasing greenhouse gas emissions over the last decades could be responsible for the warming in the Subantarctic Front[31]. In contrast, wind-driven cold and fresh water is upwelled along isopycnals[32] in the southern (cold) flank of the Polar Front (400–500 km; Fig. 6a, d); this upwelling is likely due to the SO westerly wind stress intensification[3]. The heave component shows cooling and salinification across the Polar Front in the upper 300 m, and it is maximum in the front's south flank (400–500 km; Fig. 6b, e), which makes up most of the signal at depth in the temperature and salinity trends (Fig. 6c, f). The heave-driven cooling potentially stems from a more negative wind stress curl (Ekman suction or upwelling) over the Drake Passage area during the last 15 years (Supplementary Fig. S2), consistent with previous trends estimated for a shorter period of time[33]. A similar cooling trend in sea surface temperature in Drake Passage was found from 1993 to 2017[34]. Our study finds that the cooling trends between 2005 and 2019 are a factor of two-three larger

than those in ref. 35 potentially owing to the coverage over a different period and the average area used in ref. 35. South of the Polar Front (below 150-m depth), strong heave-driven warming is associated with the upwelling of warm-salty UCDW (> 500 km; Fig. 6b, e). The upwelled UCDW below the mixed layer could minimize AWW's formation, which could produce a warmer water column.

**Trends in eddy activity**

As discussed in "Trends in transport across Drake Passage", the acceleration of the transport in the region between the Subantarctic and Polar Fronts and the deceleration of the Polar Front jet (Figs. 3 and 4) potentially arise from an increase in the steering and meandering of the ACC fronts due to enhanced mesoscale eddy activity. To examine this hypothesis, we constructed time series of depth-averaged eddy kinetic energy and eddy momentum fluxes per pair of streamlines (<EKE > and < EMF > , respectively; see "Methods").

Time series of streamwise < EKE > show that eddy energy increases during the last quarter of the sampling period (Fig. 7a) in the zone between the Subantarctic Front and Polar Front (−0.60 m > SSH > − 1.00 m). This zone exhibits the largest time-mean (denoted by overbar) depth-averaged eddy kinetic energy (<EKE>), consistent with previous studies[26–28]; <EKE> decreases significantly away from this region (Fig. 7b). Trends show a significant increase in < EKE > in the south flank of the Subantarctic Front over the last 15 years (Fig. 7c), indicating an increase in the meandering of the front's streamlines and eddy formation. Surprisingly, trends show a significant decrease of <EKE> in the Polar Front (Fig. 7c). Our means are unaltered when they are calculated using only those (60) velocity transects that have a coincident temperature/salinity transect (Fig. 7b). Trends using the coincident transects show a positive peak centered in the zone between the Subantarctic and Polar Fronts; this peak lies poleward of the significant positive trend estimated using all transects, although trends using the coincident transects are insignificant (Fig. 7c). Our

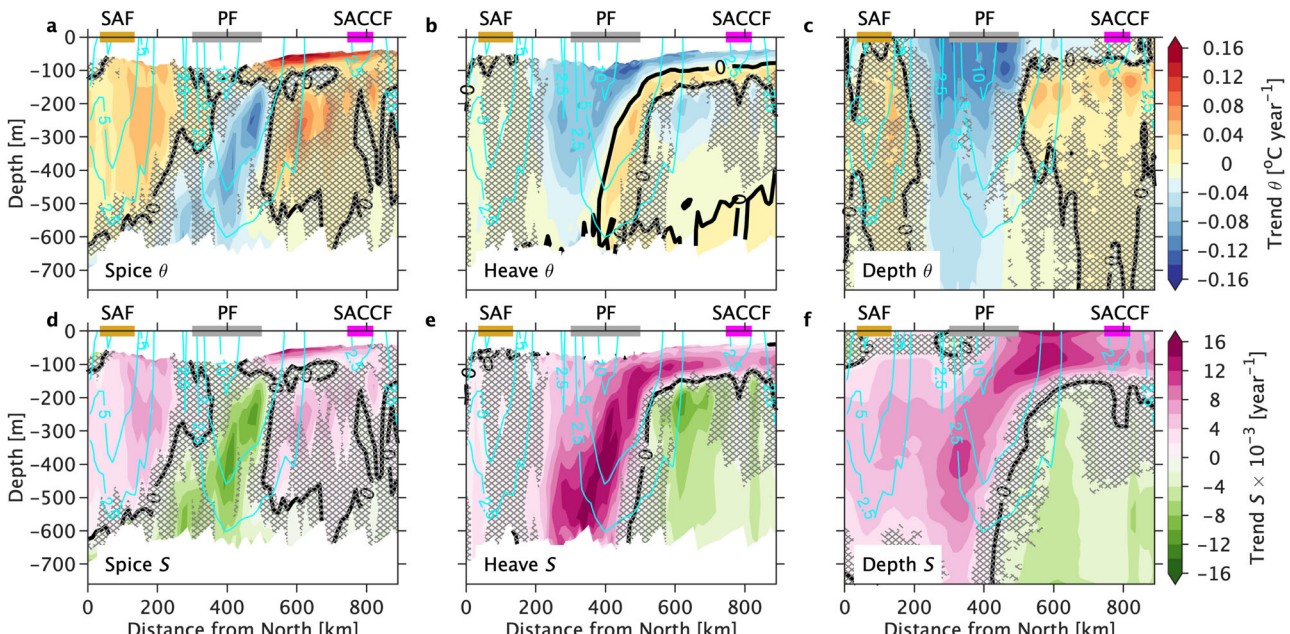

**Fig. 6 | Spice, heave, and depth trends of potential temperature [°C year⁻¹] and salinity [year⁻¹] across the Drake Passage from October 2005 to April 2019.** Trends in (**a**–**c**) potential temperature $\theta$ and **d**–**f** salinity $S$ were decomposed in (**a**, **d**) along neutral density "spice", (**b**, **e**) isopycnal heave and (**c**, **f**) depth or total trend. Positive trends indicate warming and salinity increase for $\theta$ and $S$, respectively. Black thick contour shows the zero trend. Non-hatched areas indicate that trends are statistically significant after the modified Mann−Kendall test ($p < 0.05$). Cyan contours show the 15-year mean $u_{geo}$ plotted every 2.5 cm s⁻¹. Horizontal gold, gray, and magenta bars indicate the mean location of the Subantarctic Front (SAF), Polar Front (PF), and Southern Antarctic Circumpolar Current Front (SACCF).

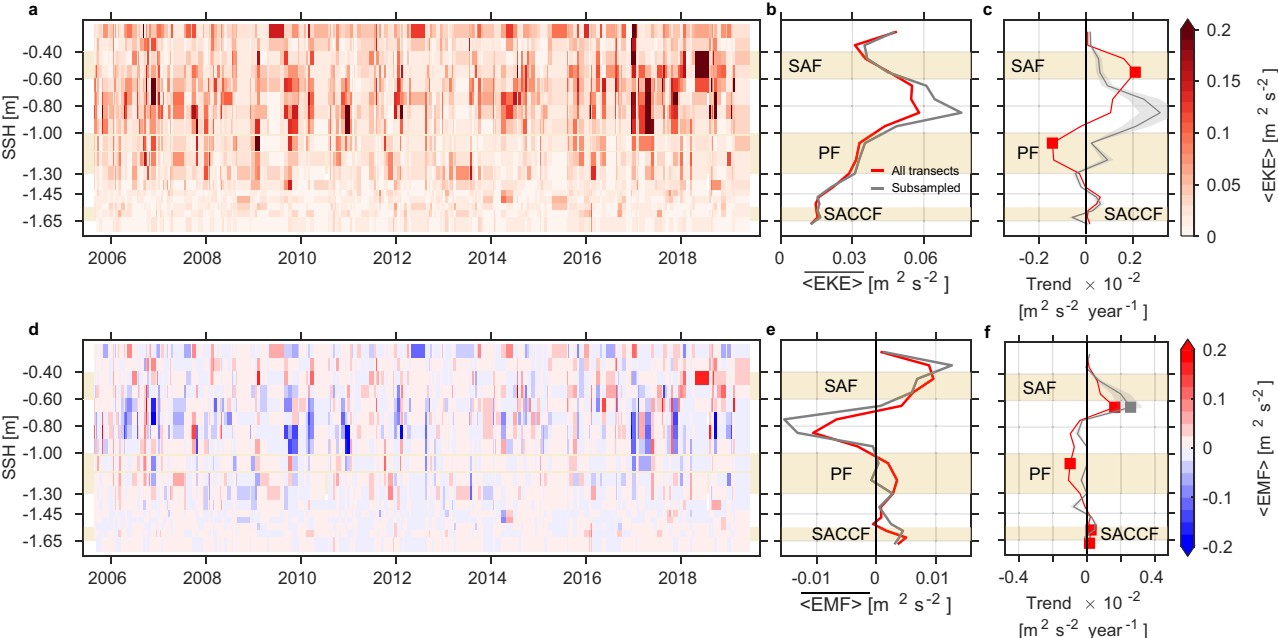

**Fig. 7 | Time series of eddy kinetic energy (EKE) and eddy momentum fluxes (EMF), mean, and trends per pair of sea surface height (SSH) streamlines [m] for October 2005–April 2019.** Time series of depth-averaged total **a** < EKE > and **d** < EMF > in the upper 760 m [m² s⁻²]. Mean **b** $\overline{\text{<EKE>}}$ and **e** $\overline{\text{<EMF>}}$ [m² s⁻²]. Trends in (**c**) < EKE > and (**f**) < EMF > [m² s⁻² year⁻¹]. Red and gray lines (**b**, **c**) show the mean and trends estimated for all (237) available transects and only those (60) velocity transects that have a concurrent temperature/salinity transect, respectively. Filled squares (**c**, **f**) indicate that trends are significant at 95% confidence after the modified Mann−Kendall test ($p < 0.05$). Standard errors for each line are shown in shaded areas in (**c**, **f**). Shaded rectangles show the streamwise intervals for the Subantarctic Front (SAF), Polar Front (PF), and Southern Antarctic Circumpolar Current Front (SACCF)[28].

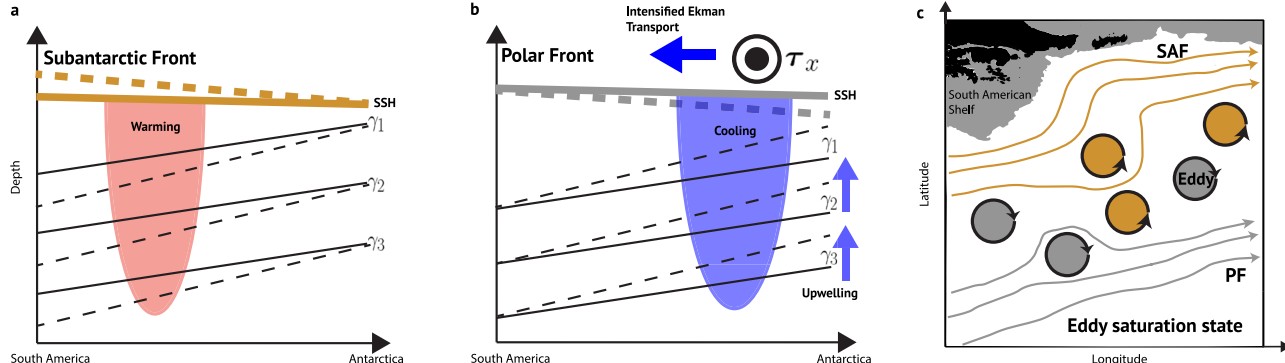

**Fig. 8 | The response of the Antarctic Circumpolar Current (ACC) to buoyancy changes within the frontal regions over the last 15 years.** Buoyancy changes across the **a** Subantarctic Front (SAF) and **b** Polar Front (PF). Red patch in (**a**) shows warming and blue patch in (**b**) shows wind-driven cooling. Blue thick horizontal arrow in (**b**) is the intensified equatorward Ekman transport due to increasing westerly wind stress $\tau_x$ (circles). Blue vertical arrows in (**b**) show the enhanced wind-driven upwelling and shoaling of isopycnals. Solid black lines are the fronts isopycnals $\gamma$ ($\gamma_1 < \gamma_2 < \gamma_3$) in normal conditions; dashed lines are steepened isopycnals due to the buoyancy changes in the fronts. Surface thick solid lines are the sea surface height (SSH) in normal conditions; thick dashed lines are the SSH of the steepened fronts' SSH. **c** Eddy saturation state in Drake Passage. Solid gold and gray contours show the SAF and PF SSH streamlines, respectively, with arrowheads showing the flow direction within the frontal jets. Filled gold and silver vortices are mesoscale eddies detached from the SAF and PF, respectively.

results are consistent with an eddy saturation state, i.e., the net ACC transport remains invariant to the increased wind-forcing whereas the eddy activity increases[9,15].

Time series of streamwise < EMF > show that the largest negative values occur in the region between the Subantarctic and Polar Fronts, whereas relatively smaller positive values dominate in the Subantarctic Front (Fig. 7d). The time-mean depth-averaged eddy momentum flux $\overline{<EMF>}$ reflects the pattern seen in the time series; it is positive in the Subantarctic Front (SSH > −0.65 m) and negative in the region between the fronts (−0.65 m > SSH > −1.00 m) (Fig. 7e). Within and south of the Polar Front (SSH < −1.00 m), $\overline{<EMF>}$ is again positive although its value is close to zero (Fig. 7e). Positive values imply a northward flux of eastward momentum and negative values imply a southward flux of eastward momentum[26]. This interpretation suggests that, on average, the eddies in the interfrontal zone redistribute zonal momentum towards the Subantarctic Front and the Polar Front. The positive trend of < EMF > in the southern flank of the Subantarctic Front (Fig. 7f) suggests that eddies have significantly increased the northward flux of zonal momentum in the south flank of the Subantarctic Front over the last 15 years. Similar results are obtained when means and trends are calculated using only those transects that have a coincident temperature/salinity transect (Fig. 7e, f).

## Discussion

Time series of along-track upper-ocean temperature, salinity, and ocean currents across Drake Passage, the narrowest constriction of the ACC, are employed to estimate trends in the Drake Passage transport and properties over a period of 15 years. The time series employed in this study have sufficient duration and horizontal resolution to diagnose the temporal trend and resolve the spatial structure of the ACC response to changing winds, something not possible using other platforms such as the sparser Argo array[31].

We found that, although the upper 760 m net total, geostrophic and reference Drake Passage transports do not show significant acceleration (Fig. 2 and Supplementary Fig. S1), consistent with previous studies[17,20,21,31], the net transport trend comprises compensating trends across the ACC frontal regions. The acceleration of the region between the Subantarctic and Polar Fronts (Fig. 3 and 4) appears to be an artifact of the geographic coordinate system. When analyzed within streamlines (Fig. 5), the pattern of acceleration/deceleration is much reduced and mostly insignificant, suggesting that the apparent acceleration in the interfrontal zone is due to an increase in meanders and

eddies. The streamwise analysis attributes meanders and eddies to the source jets, even when the deviation in location is large.

Trends of depth-averaged EKE estimated from the velocity transects show that eddy activity has increased over the last 15 years in the zone between the Subantarctic and Polar Fronts, peaking on the southern flank of the Subantarctic Front (Fig. 7c). The increased EKE is consistent with altimetry-based EKE trends observed in ACC eddy hot spots like Drake Passage over the last two decades[36,37]. Our interpretation is consistent with an eddy saturation state, i.e., eddy activity increases as the wind stress increases with no net acceleration of the ACC[9-15] and with observed buoyancy changes (Fig. 6). The warming in the Subantarctic Front (red patch, Fig. 8a) and wind-driven cooling south of the Polar Front (blue patch, Fig. 8b) act to steepen the isopycnals across the fronts (dashed tilted, Fig. 8a, b), enhancing the available potential energy stored in the fronts that is subsequently released through baroclinic instabilities[38]. Consequently, more baroclinic instabilities translate into more steering and meandering of the fronts and eddy formation (Fig. 8c), which feed into the vigorous eddy field in Drake Passage[25-28].

Moreover, the mean pattern of eddy momentum flux shows that eddies have acted to redistribute momentum across the ACC frontal regions over the last 15 years, by removing momentum from the interfrontal zone and depositing it in the jets (Fig. 7e). The positive trend in eddy momentum flux at the southern boundary of the Subantarctic Front could help to accelerate the jet (Fig. 7f). The role of eddy momentum flux in redistributing momentum due to increased winds and its effect on the ACC jets is an outcome not covered in the eddy saturation state hypothesis and warrants future work. Eddy-permitting models with sufficient resolution to resolve the ACC fronts in Drake Passage[21], forced with both realistic and idealized wind-forcing, provide an opportunity to study the potential eddy-driven sharpening of the ACC fronts posed in this study.

The increase in eddy activity over the last 15 years has implications for the SO meridional overturning circulation. In the upper cell of the overturning circulation, the increase in the northward Ekman transport is compensated by deep poleward eddy transport across the ACC fronts[1], facilitating the poleward transport of heat, salt, and tracers that play a significant role in the global ocean heat budget. Given the sustained increase in global greenhouse emissions, the westerly wind stress over the ACC is expected to continue to rise, which would act to increase the poleward eddy transport

and thus the upwelling of deep warm waters along isopycnals out-cropping near the Antarctic continental shelf [39,40]. This intensification of the deep-water upwelling is of significance for changes in the SO sea-ice interactions, ice-melting[33,41,42], global sea-level rise, and the carbon sink.

## Methods

### The Drake Passage datasets

Underway upper-ocean velocity, temperature, and salinity were collected aboard the Antarctic Research and Supply Vessel (ARSV) Laurence M. Gould (LMG) that transits between South America and the Antarctic Peninsula (Fig. 1). Since February 2005, a 38 kHz phased array acoustic Doppler current profiler (ADCP; OS38) has sampled velocity in the upper 1000 m at 24-m vertical resolution with the first depth bin at 46 m. Transects were discarded if one of the following criteria were met: (a) transects were not completed in less than 4 days, (b) along-track/vertical gaps were too large to fill, or (c) if transects with gaps were outside the area enclosed by the gridded objectively mapped mean. Following these criteria allows us to estimate the mean and trends in both the net total transport and transport per distance bin with the same number of degrees of freedom (transects). From February 2005 to December 2019, 248 (of 286 total crossings) ADCP transects met these criteria, with 147 transects along the most commonly repeated line (Supplementary Fig. S1). From October 2005 to April 2019, the common sampling period between the ADCP and the temperature/salinity transects (details below), 237 transects were available with 140 transects falling along the most repeated line.

Velocity data were processed using the Common Ocean Data Access System (CODAS) software[43]. Following refs. 24,44, returned ping data were averaged over 5-min (100 pings) ensembles and screened using amplitude, error velocity, and percent good criteria. Velocities were transformed from ship-relative to absolute ocean currents using GPS position and attitude measurements. The absolute current velocities were further averaged to 15-min resolution, i.e., ~4.50 km along-track resolution (assuming a ship's speed of 5 m s$^{-1}$). Barotropic tidal currents were removed from the absolute velocity by subtracting the tidal prediction of the TPXO7.2 tide model[45]. A slab layer was assumed for the upper 46 m where the OS38 ADCP did not sample. The single ping velocity uncertainty is 23 cm s$^{-1}$; thus the uncertainty of a 300 ping average is 1.3 cm s$^{-1}$. Combined with the error in ship speed from GPS (0.8 cm s$^{-1}$) the error in absolute velocity in a 15-min average is 1.5 cm s$^{-1}$. Averaging over 83 min (i.e., ~25 km along-track assuming an average ship speed of 5 m s$^{-1}$) reduces the error $\sigma_{tot}$ to 0.65 cm s$^{-1}$.

Since February 1997, on 6–7 LMG transects per year ~70 eXpendable Bathythermograph (XBT) probes are deployed that measure temperature in the upper 900 m. The spatial resolution of the XBT sampling is 6–10 km across the Subantarctic Front and Polar Front, and 10–15 km elsewhere[46]. All data were quality-controlled and corrected as in ref. 46. The error associated with the systematic fall rate error is ~1% of the depth due to random XBT probe differences. The thermistor calibration error is 0.05 °C. The majority of crossings also deployed twelve expendable conductivity–temperature–depth (XCTD) probes that directly measure temperature and salinity in the upper 1000 m with a sample spacing of 25–50 km. The XCTD probe salinity accuracy of ~0.05 accounts for the errors in conductivity, temperature, and pressure[47]. To obtain cross-transect salinity, a look-up temperature-salinity-depth-position relationship was first constructed from historical hydrography. For each XBT temperature profile, a corresponding salinity was obtained. The salinity anomaly was then determined from the XCTD measurements with respect to the historical data, and the salinity anomaly was objectively mapped along the transect and added to the corresponding XBT-derived salinities. Overall, there were 114 XBT/XCTD (temperature/salinity) transects from February 1997 to

December 2019 that were used to calculate the time-mean temperature and salinity fields; 53 of these transects fall along the most commonly sampled line (Supplementary Fig. S1). From October 2005 to December 2019, 69 of 114 transects were available of which 60 transects coincide with the OS38 ADCP transects.

### Objective mapping temperature and salinity transects

Each temperature and salinity transect was objectively mapped to a latitude-depth grid with a horizontal and vertical resolution of 1/10° and 10 m starting from the surface to 760 m, respectively, a depth range commonly reached by most XBT and XCTD probes. Gaps in the XBT/XCTD data below 760 m were relatively large, therefore, making the normalized mapping error larger than the noise-to-signal ratio. For the temperature horizontal grid, two Gaussian functions were used: one with a large-scale decorrelation scale of 100 km and another with a scale of 30 km. Likewise, two scales were employed for the XCTD salinity: a 100-km scale and a 50-km scale. The latter was employed since the spacing between XCTD profiles is larger than that for the XBT profiles.

### Constructing time series of total velocity

The velocity vector was rotated to an along/cross-transect coordinate system. For this study, only the cross-transect component $u_{tot}$ was kept (positive flow is eastward) and then bin-averaged onto a 25-km along-track grid[24]. For transects with continuous gaps ≤150 km, the gaps were filled using an objective mapping as follows. Velocity anomalies were estimated by subtracting the time-mean cross-transect velocity component from the $u_{tot}$. The time-mean velocity is obtained by calculating objectively mapped mean velocity vectors as in ref. 28 (and references therein). In calculating the time-mean velocity, velocity transects from all 286 crossings were employed. For each transect, the nearest time-mean velocity (in a 25 km × 25 km grid-box) to the $u_{tot}$ velocity profile was employed and rotated to the along/cross-transect coordinates to estimate the time-mean cross-transect velocity component and subtract it from $u_{tot}$. Next, the total velocity anomalies were objectively mapped into an along-track/depth grid using a Gaussian covariance function with horizontal and vertical decorrelation scales of 50 km and 300 m, respectively. Subsequently, the gaps were filled with the mapped anomalies plus the time-mean cross-transect component. Finally, the OS38 time series were corrected following ref. 24 due to a systematic offset between northbound and southbound Drake Passage total transport. The transport bias arises from a bias in the cross-transect velocity and is consistent with a rotation calibration error due to uncertainty in the measured transducer alignment. This angle error results in a component of the ship's speed erroneously projected in the cross-track component of velocity. Supplementary information provides details about the method employed to minimize transport bias.

### Total, geostrophic, and reference transport

The upper-ocean net total Drake Passage transport was calculated as $U_{tot} = \int_0^L \int_{z_0}^0 u_{tot} dz dx$, where $L$ is the length of the transect, $dx$ is the horizontal spacing (25 km), and $z_0 = -760$ m is the common maximum depth between the OS38 and XBT/XCTD profiles. The error in absolute transport in the upper 760 m per 25-km along-track bin ($dx$) is estimated from the absolute velocity error ($\sigma_{tot}$) as $\sigma_{tot} * dx * H/\sqrt{n}$, where $H = 760$ m and $n$ is the number of observations (transects), yielding $8 \times 10^{-3}$ Sv for 237 transects for the October 2005–April 2019 period. Similarly, the standard error in absolute net Drake Passage total transport, estimated as $\sigma_{tot} * L * H/\sqrt{n}$, is 0.28 Sv.

The geostrophic cross-transect velocity component $u_{geo}$ and Drake Passage transport $U_{geo}$ were estimated from the objectively mapped temperature and salinity transects. The transects were employed to estimate the geopotential anomaly $\Phi$, which is the

geostrophic streamfunction, given as:

$$\Phi = \int_{p_0}^{p} \delta(x,p)dp, \qquad (1)$$

where $p_0 = 760$ db is the reference pressure, $p$ is pressure, and $\delta$ is the specific volume anomaly estimated from the temperature and salinity transects. Cross-transect geostrophic velocity $u_{geo}$ was then calculated as the first difference of $\Phi$ with respect to the along-track distance:

$$u_{geo} = -\frac{1}{f}\frac{\partial \Phi}{\partial x}, \qquad (2)$$

where $f$ is the local Coriolis frequency, and $x$ is the along-track distance. The geostrophic Drake Passage transport was calculated as $U_{geo} = \int_0^L \int_{z_0}^0 u_{geo} dz dx$ ($dx \sim 12$ km).

The total cross-transect velocity $u_{tot}$, defined as that of the ADCP velocities, can be decomposed into geostrophic $u_{geo}$, reference $u_{ref}$ and Ekman $u_{Ekm}$ components (i.e., $u_{tot} = u_{geo} + u_{ref} + u_{Ekm}$) when XBT/XCTD transects are available. We estimated the Ekman component by subtracting the geostrophic component profile from the total component profile [i.e., $u_{Ekm} = u_{tot} - u_{geo}$; ref. 48] within the Ekman layer ($z \leq z_{Ekm}$). The Ekman layer was defined where the total shear showed an exponentially decaying profile, after first averaging per transect and then averaging for the entire time series[48]. For our observations, the base of the Ekman layer was $z_{Ekm} = -90$ m since the mean shear velocity reduces to a constant value below 90 m and is in good agreement with the geostrophic shear. Here, we interpolated the total velocity to the geostrophic velocity along-track and depth grid for each transect. Thus the Ekman transport was calculated as $U_{Ekm} = \int_0^L \int_{z_{Ekm}}^0 u_{Ekm} dz dx$. Consequently, the reference component $u_{ref}$ was defined as the averaged residual velocity between the base of the Ekman layer and the deepest bin $z_0$:

$$u_{ref} = \frac{1}{|z_{Ekm} - z_0|}\int_{z_0}^{z_{Ekm}} (u_{tot} - u_{geo})dz. \qquad (3)$$

Therefore, the reference Drake Passage transport is $U_{ref} = \int_0^L \int_{z_0}^{z_{Ekm}} u_{ref} dz dx$.

## Isopycnal heaving and spiciness

To understand the changes in potential temperature $\theta$ and salinity $S$ with depth and position across Drake Passage, temperature and salinity trends were decomposed into two main contributions[29]: modification of temperature and salinity along isopycnals (neutral density layers $\gamma$) called "spiciness", and vertical displacement "heave" of neutral density layers[49]. As discussed in ref. 30, the analysis when done along pressure-density surfaces allows a direct calculation of the component of temperature and salinity change due to vertical heave of neutral density surfaces, and that due to spice. The decomposition of the total potential temperature and salinity changes $\frac{\partial \theta}{\partial t_p}$ and $\frac{\partial S}{\partial t_p}$ on a pressure surface is:

$$\overline{\frac{\partial \theta}{\partial t_p}} = \frac{\partial \theta}{\partial t_\gamma} + \frac{\partial \overline{\theta}}{\partial \gamma}\frac{\partial \gamma}{\partial t_p} + \text{Res}, \qquad (4)$$

$$\overline{\frac{\partial S}{\partial t_p}} = \frac{\partial S}{\partial t_\gamma} + \frac{\partial \overline{S}}{\partial \gamma}\frac{\partial \gamma}{\partial t_p} + \text{Res}. \qquad (5)$$

The first terms on the right-hand side of Eqs. (4) and (5) represent the changes on neutral density levels, whereas $\frac{\partial \overline{\theta}}{\partial \gamma}$ and $\frac{\partial \overline{S}}{\partial \gamma}$ are the local gradient of mean $\theta$ and $S$ in $\gamma$, and $\frac{\partial \gamma}{\partial t_p}$ represents the neutral density change on a pressure surface. Res represents other terms from the Taylor expansion that are considered as a residual.

An important point is that the spiciness changes represent a shift in the $\theta/S$ profiles at a constant $\gamma$. Consequently, a change in salinity involves a change in temperature and conversely, temperature change along $\gamma$ involves a change in salinity. The heave component reflects adiabatic processes such as wind-driven Ekman pumping and low-frequency Rossby waves.

## Tracking the position of the ACC streamlines

To determine the trends across Drake Passage as a function of the ACC frontal regions, we employed a synoptic streamwise coordinate system (sea surface height; SSH) in a similar way as used in ref. 28 for eddy heat flux. We used a combination of the mean dynamic topography[50] and the SSALTO/DUACS daily maps of sea-level anomalies from AVISO to track the position of the ACC streamlines. The mean dynamic topography is directly tied to the large-scale surface geostrophic circulation constrained by 20 years of in situ observations such as drifters corrected due to wind/Ekman currents, altimetry, Argo temperature/salinity profiles, and conductivity–temperature–depth (CTD) casts[50]. The daily maps were obtained from multiple satellite altimeters and objectively mapped to a $0.25° \times 0.25°$ Cartesian grid[51]. The sea-level anomalies are relative to a 20-year mean of the sea surface height field. Adding the mean dynamic topography to the sea-level anomalies produces maps of dynamical SSH that enable tracking of the shifting and meandering of the ACC streamlines and mesoscale eddies. We used the daily maps from February 1999 to December 2019, which covers our period of interest, and we subtracted a 0.6 m constant from the daily maps following ref. 28 (and references therein), which does not affect the position of the fronts and their horizontal gradients.

The daily SSH are smoothed with a 20-day box-car window; we tested if the binning, and thus the trends, are sensitive to the smoothing window, however, no significant changes in the trends were observed. The tracking method provides an algorithm to attribute and bin data on occasions when a transect crosses the same front more than once. For example, transects often cross the Subantarctic Front twice, as it turns steeply from eastward to northward following the South American shelf break, forming meanders that can detach as eddies with closed contours. Consequently, more profiles fall within closed contours when the fronts meander or during eddy formation and detachment (see Appendix in ref. 28). Hence, we expect that an increase in meandering and eddy formation augments the number of transects that cross a meander or eddy. This aspect of the synoptic SSH method could impact our interpretation of the acceleration of the jets on the ACC fronts. Velocity profiles falling within each unique pair of SSH streamlines are integrated vertically with depth and distance to form a transport estimate per pair of streamlines per transect. $u_{tot}$ is re-interpolated to a ~12 km along-track distance (similar to that of the $u_{geo}$ and $u_{ref}$) so that the criterion of one point per pair of streamlines per transect is met.

## Estimating eddy kinetic energy and eddy momentum flux

To quantify trends in the eddy activity in Drake Passage and the eddy momentum transfer with the ACC jets, we estimated time series of depth-averaged eddy kinetic energy and eddy momentum flux from the 5-min (100 pings; 1.5 km along-track resolution) OS38 ADCP records corrected for transducer misalignment angle (see supplementary information Eq. (4)). We focus on the period of October 2005–April 2019 (237 velocity transects), which is the period employed to estimate trends in cross-transect velocity and streamwise transport. Following[28], eddy velocity terms per transect per depth bin were estimated as $\mathbf{u}' = \mathbf{u} - \overline{\mathbf{u}}$, where $\mathbf{u}$ and $\overline{\mathbf{u}}$ are the 5-min velocity vector and the time-mean velocity vector. Subsequently, depth-averaged (0–760 m) eddy kinetic energy (EKE) and eddy momentum

flux (EMF) per transect were calculated as

$$EKE = \frac{1}{H} \int_{z_0}^{0} 0.5 * (u'^2 + v'^2) dz, \qquad (6)$$

$$EMF = \frac{1}{H} \int_{z_0}^{0} u'v' dz. \qquad (7)$$

Next, each EKE and EMF transect was bin-averaged at a ~12 km along-track distance and then streamwise averaged using the synoptic SSH to construct time series of $<EKE>$ and $<EMF>$ per transect per pair of streamlines, where the angle brackets $< \cdot >$ denote the bin and streamwise averaging. The ~12 km resolution allows us to meet the criterion of at least one point per pair of streamlines per transect.

### Statistical analysis: trends and significance

Trends in Drake Passage transports ($U_{tot}$, $U_{geo}$, and $U_{ref}$), cross-transect velocities ($u_{tot}$, $u_{geo}$, and $u_{ref}$), salinity $S$, potential temperature $\theta$, $<EKE>$ and $<EMF>$ were calculated after removing the seasonal cycle (sum of the annual and semiannual harmonics). The trends were estimated by least-squares fitting $\hat{y} = a_0 + a_1 t$ to each time series; the second term $a_1$ on the right-hand side of the equation is the trend. Uncertainty in the calculated least-square trends is reported as the 95% confidence intervals[52] using

$$CI_{a_1} = \pm t_{\alpha/2} \times RMSE \times \sqrt{\frac{1}{\sum (t_i - \bar{t})^2}}, \qquad (8)$$

$$RMSE = \sqrt{\frac{\sum (\hat{y} - y_i)^2}{n - \mu}}, \qquad (9)$$

where $t_{\alpha/2}$ is the Student's t statistic for 95% confidence (i.e., $\alpha = 0.05$), RMSE is the root mean square error, $\hat{y}$ is the predicted variable, $y_i$ is the observed variable, $n - \mu$ is the number of observations minus the number of parameters fitted $\mu = 2$, and $\bar{t}$ is the average time in years.

Statistical significance of the trends employed a modified Mann–Kendall test[53], a commonly used nonparametric trend test. The modified test takes into account autocorrelations within the time series. The variance of a variable for autocorrelated data can be calculated as

$$V^* = \sigma^2 \cdot \frac{n}{n^*} = \frac{n(n-1)(2n+5)}{18} \cdot \frac{n}{n^*}, \qquad (10)$$

$$n/n^* = 1 + \frac{2}{n(n-1)(n-2)} \times \sum_{i=1}^{n-1} (n-i)(n-i-1)(n-i-2)C(i), \qquad (11)$$

where $n/n^*$ represents a correction related to autocorrelation in the time series, and $C(i)$ denotes the autocorrelation between the ranks of observations. This study assumes that trends were statistically significant when the modified Mann–Kendall test resulted in a probability $p < 0.05$.

### Data availability

XBT temperature data are available at the Scripps High-Resolution XBT program website (http://www-hrx.ucsd.edu). ADCP time series were acquired and processed by the Chereskin Lab at Scripps Institution of Oceanography (http://adcp.ucsd.edu/lmgould/) and can be downloaded from the Joint Archive for Shipboard ADCP (JASADCP) (https://uhslc.soest.hawaii.edu/sadcp/). The SSALTO/DUACS altimeter products were produced and distributed by the Copernicus Marine and Environment Monitoring Service (CMEMS) (http://www.marine.copernicus.eu). The mean dynamic topography[50] for the period 1992-2012 is available at http://apdrc.soest.hawaii.edu/projects/DOT. The ERA-5 Reanalysis[54] was produced by the European Center for Medium-Range Weather Forecast (ECMWF) and downloaded from the Copernicus Climate Change Service (C3S) Climate Data Store (2022) and is available at https://cds.climate.copernicus.eu/cdsapp#!/dataset/reanalysis-era5-single-levels?tab=form.

### Code availability

MATLAB code for the analysis and plotting figures for the main manuscript and supplementary information is available at https://github.com/manuelogtzv/TrendsDrakePassageTransp[55]. Datasets used to estimate time series used for estimating trends are available at https://doi.org/10.5281/zenodo.10044261.

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

## Acknowledgements

M.O.G.V., T.K.C., and J.S. acknowledge the National Science Foundation's Office of Polar Programs (OPP) and Division of Ocean

Sciences (OCE) for support of the Drake Passage time series and this research through grants OPP-2001646 and OCE-1755529. The XBT probes are provided by NOAA's Global Ocean Monitoring and Observing Program through Award NA20OAR4320278 to J.S. M.O.G.V. was supported by the University of California Institute for México and the United States (UC Mexus) and the Consejo Nacional de Ciencia y Tecnologia of México (Conacyt) Doctoral fellowship (361655). Finally, M.O.G.V., T.K.C., and J.S. are also grateful to the captain and crew of the ARSV *Laurence M. Gould* and the Antarctic Support Contractor for their excellent technical and logistical support.

## Author contributions

M.O.G.V. analyzed the data and wrote the paper. T.K.C. supervised data collection and processing and assembled the ADCP data for analysis. J.S. supervised data collection and processing and calculated the historical look-up table and objectively mapped the XBT/XCTD data for analysis. M.O.G.V., T.K.C., and J.S. designed the research, interpreted the results, and made revisions to the paper.

## Competing interests

The authors declare no competing interests.
