## [Peer Review File · Nature Communications]

Compensating transport trends in the Drake Passage frontal regions yield no acceleration in net transportREVIEWER COMMENTS

Reviewer #1 (Remarks to the Author):

In “compensating transport trends in the Drake Passage frontal regions yield no acceleration in net transport”, Gutierrez-Villanueva et al. presented the Drake Passage transport calculated from 15 years of direct measurements (2005-2019) for the top 760 m, based on shipboard ADCP surveys along with XBT/XCTD measurements (that covered from 1997 to 2019). The key finding is that while the net transport measured exhibits no acceleration, the key front regions do exhibit statistically significant trends that have opposite signs. They further suggested that the acceleration of the frontal jets results from an increase in the mesoscale eddy activity due to buoyancy changes in the fronts, consistent with an eddy saturation state. I believe this observational work is of great value to the community and, to some degree, I think the authors can emphasize a bit more on the strength and/or the importance of this unique observation, i.e., high-resolution direct measurement of the current/transport across the full passage (for the upper ocean). However, there are some places I think the point is a bit unclear or potentially misleading, hence some clarifications are needed.

1. The compensating trends in frontal regions “explain” (L20) or “are likely responsible for” (L69) the lack of trend in the net transport. Probably a minor wording issue but I am not sure if this causative implication is appropriate. The net transport is made of the transports of the front regions, so it is somewhat strange to call one explains/is responsible for the other. Simply stating the results, i.e., no trends in the net transport, but compensating trends in the frontal regions, is more appropriate to me.

2. I do not think it is accurate to state “SAF and PF jets have accelerated while the currents located between the front have decelerated ...” (L17-18). If one reads Fig. 3 correctly, the strongest increase is located between SAF and PF, the southern half of the PF and the region between PF and SACCF show decreasing velocity (L130-134; L185-186).

3. Not sure if “acceleration of frontal jets results from an increase of mesoscale eddy activity” is consistent with eddy saturation state (L22-23). If an increase of mesoscale eddy activity leads to acceleration of the frontal jets, it should ultimately lead to a stronger net ACC transport, right? And what contributes the decreasing trend in the southern part of PF and north of SACCF?

4. I wonder if it helps to remove the mean from Fig 5, because it is difficult to see trend (which might be the reason to have Fig. 6). More importantly, I think it is valuable to have a similar time series plot from the geophysical coordinate perspective, showing over time the change of vertically integrated velocity that corresponds to the trend in Fig. 3 (This plot can even include those sections that did not cross 1000 m isobath, if those sections have been processed, just not included in transport calculation).

5. The ADCP based transport is calculated for the top 760 m, not to 1000 m as in the previous work covering 4.5 years of the observations (Firing et al., 2011). I assume this is because many sections do not extend to 1000 m, otherwise, one may argue it is valuable to have a separate time series that cover the

top 1000 m to be consistent with the previous work (separating between geostrophic and reference transport at 760 m seems of less importance to me, although that is arguable).

Other details:

L39. Ref 4 may be placed at the end of this sentence as it is about both ozone and greenhouse gas emissions (It should be mentioned that it is a modeling work). Not sure why ref 5 (also a modeling work) is included here as it is mostly about ocean warming, not so much about wind or southern annular mode.

L58, "... time series were not yet long enough" and L111. "... using short sampling periods" are not accurate for refs. 16 and 19. The geostrophic transports based on hydrographic surveys (ref 16) cover a pretty long period (although less frequent). Ref 19 rely much on satellite SSH which is not direct measurement of current. I think the strength or value of this 'unique' observation needs to be highlighted a bit more, i.e., high-resolution, direct measurements of the current across the full Drake passage over a long period of time (One limitation is that it covers only upper part of the water column).

L113. Refs. 21, 22 find no acceleration in the Drake passage but their main result is that the warming is leading to acceleration in zonal flow in much of the Southern Ocean, and I do not think low-resolution climate model is the best examples to show no acceleration in ACC transport (because without eddies, a stronger wind tends to drive a stronger ACC transport).

L114-L120. Ref. 23 found that "baroclinic and barotropic transports are not correlated; thus, monitoring either baroclinic or barotropic transport alone may be insufficient to assess the temporal variability of the total ACC transport". They did not imply that the lack of trend in their modeled transport is due to two components compensating each other. Furthermore, the separation of geostrophic and reference velocity at 760 m has different meaning from the baroclinic - barotropic separation for full water column. On a different perspective, I think it is noteworthy that ref 23 carefully evaluated the modeled ACC transport structure using various observations, including the result of the first 4.5 years of the ADCP measurement that are presented in this paper. The agreement between model and the ADCP measurement for the top 1000 m was one key reason that gives the confidence in their modeled transport. Thus, the observations like this are also of great value to modeling community as well. I think this point could be noted somewhere in the paper.

L176. "[32] found a similar ..." you may rephrase this as "a similar cooling ... [32]".

Fig.1 I assume panels b-d is along the common section? If yes, it should be mentioned in the Figure caption and noted that the location of this section is indicated in S1, or add a one line in panel a.

Reviewer #2 (Remarks to the Author):

This paper uses a 15-year time series of subsurface temperature, salinity, and velocity in the Drake passage to investigate the structure of the transport trends of the Antarctic Circumpolar Current. The paper presents the trends in the overall net transport across the Drake Passage, cross-passage velocity and streamwise transport, and link them with the buoyancy changes across the passage.

These results hold significant importance as they offer a detailed description of the transport trends in the Drake Passage, particularly in relation to the ACC's response to strengthening winds, which remains an active research question. It is worth noting that although the Drake Passage is the most regularly sampled section of the Southern Ocean, long-term time series of temperature, salinity, and velocity across the entire Southern Ocean are scarce and highly valuable.

The paper is well written and only a few minor issues need to be addressed or clarified in my opinion.

Comments / Questions:

The paper frequently refers to the impact of wind on buoyancy and velocity change. Have the authors attempted to compute trends in winds or wind stress curl along the transect? If so, how consistent are these trends with Figure 4 and the changes described in Section 4?

L144: "is located"?

Figure 3 and Section 3 may confuse readers. While Figure 3 sets the methodology for separating the total, reference, and geostrophic components, and shows in a simple way that the compensation of the ACC frontal regions, it presents results that, at first glance, seem to contradict the final conclusions. Specifically, it shows a deceleration of velocities in the SAF and PF zones, as well as an acceleration in the region between them, whereas the abstract mentions that the subantarctic and polar jets have accelerated while the currents between them have decelerated. While this might be confusing at a first read, this confusion could be partly avoided by adding a few similar words as the opening of section 5. I leave this decision up to the authors.

L173-175: The difference in magnitude between Fig. 4 and Fig. S2 is surprising. Does that question the robustness of the trends?

L283: "to rise"

L299: Is there any specific criterion used to discard the transects that significantly deviated from the main course?

L338-340: How many of the transects between October 2005 and December 2019 fall into the most sampled line? I am just wondering how the zonal spreading of the transects could impact the cross-transect velocity, temperature and salinity trends. More specifically, can that explain part of the differences in the trends from the longer and shorter time periods (Fig. 3a vs Fig. 3b; Fig. 4 vs Fig. S2).

L425-428: What is the motivation behind the use of two different datasets sources for the Mean Dynamic Topography and Sea Level Anomaly? This could introduce discrepancies between the datasets that could be avoided with the use of the full SSH signal from SSALTO/DUACS.

L447-449: Does that mean that profiles falling into closed contours are discarded?

L459: You might mean a_0 instead of a_1

Figure 3. caption: "Trends in (a) and (b) are estimated using all available (237) velocity transects and only those (63) velocity transects that have a coincident temperature/salinity transect used to compute the geostrophic ugeo component in (c), respectively."

Please split this sentence into two parts: Trends in (a)... while trends in (b)...

Reviewer #3 (Remarks to the Author):

This paper analyses a 15-year time series of Drake Passage made up of underway ADCP measurements and XBT/XCTD deployments. The data has been collected during crossings of ARSV Laurence M. Gould during transits from South America to the Antarctic Peninsula. The measurements cover the top 1000 m, or so, of the water column with analysis restricted to between depths of 20-40 m and 760 m, due to instrument constraints and quality control processing. The resulting transects clearly show the 3 main jets of the Antarctic Circumpolar Current (ACC) and the steeply inclined isopycnals that accompany them. Analysis of least square trends shows that there are no meaningful trends in the Drake Passage transport, consistent with previous publications. However, there are statistically significant trends in the cross-transect velocity, which compensate when the area-integrated cross-transect is calculated. The locations of these trends indicates a statistically significant acceleration of the Polar Front and Subantarctic Front, with compensating deceleration between the fronts. This is interpreted as being a result of increased wind stress leading to locally steepened isopycnals and increased Eddy Kinetic Energy (EKE), with the more vigorous eddy field then sharpening and accelerating the jets due to their momentum transport.

My expertise is not in analysing and processing observational data. This makes it difficult for me to comment on these aspects of the paper. The methods section appears thorough and well thought out. It is supported by citations to previous applications of similar techniques with similar corrections, etc, being found necessary in the current paper. The statistical tests are consistently applied and systematically reported. With my inexpert eye, this all seems quite reasonable.

The paper makes a valuable contribution to understanding how the Southern Ocean and ACC has and/or will respond to climate change. Their results are consistent with previous ones showing no increase in ACC transport. However, they are able to add the important detail that this may be because of compensation between different regions increasing/decreasing their velocity. The authors link this to eddy-driving of the jets via eddy momentum fluxes and provide quantification of the same.

There are some points missing from the discussion that I think should be included. The observations only cover the top 760 m of the water column, amounting to 61% of the baroclinic transport or 47% of the transport including the near-bottom flow (lines 202-204). However, there is no discussion of whether the results are likely to hold over the whole water column. Is it likely that the same pattern of spatially localised accelerations and decelerations hold? The second thing that might warrant some discussion is whether the conclusions would hold upstream/downstream of Drake Passage. A recent paper, Shi et al. (2021), indicates that there might be acceleration of the northern edge of the ACC elsewhere in the Southern Ocean. Can the authors results shed any light on whether this might affect the circumpolar transport of the ACC? Or is this too speculative, given limited Southern Ocean observations?

Minor Comments

There were a few minor things I noticed whilst reading the paper that the authors may wish to consider.

Lines 73-76 : This sentence seems incomplete, did the authors mean to point towards Figure 1?

Figure 5b : I found it hard to tell the lines apart, because there colour is the same, even with one having prominent markers. Different colours might be better.

Lines 148-151 & Figure 3b : I was puzzled as to the difference between Figure 3a and 3b until I got to this point. It might be better to put this information in the figure caption or move it earlier in the paragraph, since readers may look at the figure before reading the full paragraph (as I did).

Response to Reviewers' Comments

NCOMMS-23-17942-T

Manuel O. Gutierrez-Villanueva^a, Teresa K. Chereskin, and Janet Sprintall

^amog002@ucsd.edu

September 26, 2023

Thank you for reviewing our manuscript. Below, we have provided detailed point-by-point replies. We have used ***bold and italic font for the Reviewers' comments*** whereas our responses are written in normal font. In our responses, we used **text in blue** to indicate that text has been incorporated into the revised manuscript. **Text in red** in the accompanied manuscript with track changes indicates that text has been removed from the original manuscript.

Major changes

Reply: We wish to mention major changes in the manuscript in this revision in response to the reviewer comments.

1. Revised Figure 2 and supplementary Fig. S1

Panel b: We found 6 additional XBT/XCTD transects for the geostrophic transport time series and trend calculations shown in Fig. 2b. Although the trend is slightly altered, the conclusion (no significant trend in the Drake Passage geostrophic transport) did not change.

Panel c: We found a mistake in calculating the reference transport time series that biased it low. We corrected the mistake and recalculated the reference transport and trend (also shown in supplementary Fig. S1). However, the conclusion (no significant trend in the Drake Passage net transport) is unaltered.

Fig. 2 Time series and trends of the Drake Passage transport [Sv] in the upper 760 m. Time series of net cross-transect (a) total U_{tot} , (b) geostrophic U_{geo} , and (c) reference U_{ref} Drake Passage transport ($1 \text{ Sv} = 10^6 \text{ m}^3 \text{ s}^{-1}$). The seasonal cycle (annual + semiannual harmonics) was removed for each time series. Standard deviations for each calendar year are shown by shading. Thick solid lines show the linear fits for the time series. Gray solid line in (b) is the linear fit for the period of October 2005 - April 2019, coinciding with (c) reference transport period. Least-squares trends and 95% error bars are shown with text [Sv year^{-1}]. None of the trends are statistically significant after the modified Mann-Kendall test ($p > 0.05$).

Fig. S1 Trends of net total (red), geostrophic (black), and reference (blue) Drake Passage transport [Sv year⁻¹]. Triangles indicate that all available transects are employed. Trends estimated for the period of October 2005 - April 2019 are marked with squares. Red circle and diamond show the trends estimated with those velocity transects that have a coincident temperature/salinity (XBT/XCTD) transect (subsampled) and along the most repeated line (red thick line in small inset; see Methods), respectively. None of the trends are statistically significant after the modified Mann-Kendall test ($p > 0.05$).

2. Revised Figure 3

Panel d: We removed this panel, as the reference velocity is constant (i.e., depth-independent) and the content is very similar to the trend in reference transport shown in the new Fig. 4i. Text in section 3 regarding the trends in cross-passage velocity has been modified to reflect the revised Fig. 3. The revised figure is shown below, and the revised text reads:

Lines 126-145: We investigated whether there are opposing regional trends in the ACC frontal jets consistent with no trend in Drake Passage transport. We estimated trends in the cross-transect total u_{tot} and geostrophic u_{geo} velocity components for the common sampling period October 2005-April 2019.

Significant opposing trends in cross-track velocity are found within Drake Passage. All significant trends ($p < 0.05$) show no change in sign with depth (Fig. 3). Since the jets of the ACC are equivalent barotropic (velocity is self-similar with depth; Killworth and Hughes, 2002), the sign of the trends likely holds over the entire water column, although the magnitude may decay. The largest significant acceleration of the eastward flow is found in the region between the Subantarctic Front and Polar Front for the total and geostrophic components (150-300 km, Fig. 3a,c). A weak, significant acceleration is found also in the Southern ACC Front (> 750 km, Fig. 3a,c). In contrast, a significant deceleration of the total flow component is found within the Polar Front (400-500 km, Fig. 3a) and to the south (550-600 km, Fig. 3a). A broader and weaker deceleration is found in the geostrophic component, but mainly within the Polar Front (350-500 km, Fig. 3c). A similar distribution in space and sign is found (although with slightly larger amplitudes) when total velocity trends are calculated with the subset of the total velocity transects that coincide with the temperature/salinity transects (Fig. 3b).

Fig. 3 Trends in cross-transect velocity components [$\text{cm s}^{-1} \text{ year}^{-1}$] from October 2005 - April 2019. Trends in cross-transect (a)-(b) total u_{tot} , and (c) geostrophic u_{geo} component. Trends in (a) are estimated using all available (237) velocity transects while trends in (b) are calculated using only those (60) velocity transects that have a concurrent temperature/salinity transect used to compute the geostrophic u_{geo} component in (c). Black thick solid contours show the zero trend. Hatched areas are non-significant trends at 95% confidence after the modified Mann-Kendall test ($p > 0.05$). Cyan contours show the mean velocity component [cm s^{-1}]. The mean location of the Subantarctic Front (SAF), Polar Front (PF), and Southern ACC Front (SACCF) are indicated by horizontal gold, gray, and magenta bars respectively.

3. New Figure 4

In responding to reviewer 1, we included a new figure showing the time series of transport anomalies and the means and trends of transport, binned by distance across Drake Passage (Fig. 4) for comparison with the same calculation made binning by streamlines (revised Fig. 5, details below). The revised figure is shown below, and the revised text in section 3 reads:

Lines 146-174: We estimated time series of cross-transect total u_{tot} , geostrophic u_{geo} , and reference u_{ref} transport per distance bin. The net total, geostrophic and reference Drake Passage transports are 75.54 ± 0.64 Sv, 15.85 ± 0.20 Sv, and 51.08 ± 0.85 Sv respectively (Fig. 4b,e,h). The sum of the geostrophic, reference and Ekman (8.09 ± 0.22 Sv; see Methods) transports yields 75.02 ± 0.90 Sv, which agrees with the net total transport calculated using the concurrent transects (77.87 ± 1.04 Sv; green line in Fig. 4b). Extending the total transport calculation to the upper 970 m yields a 15-year mean of 89.41 ± 4.69 Sv, in close agreement with Firing et al. (2011)'s mean of 95.00 ± 2.10 Sv based on the first five years of data (only using transects along the most repeated line), and the 40-year mean modeled transport of 92.7 Sv in the upper 1000 m calculated by Xu et al. (2020) from a data-constrained eddy-permitting ocean model. The 75.54 Sv net total transport in the upper 760 m is 56% of the full-depth total canonical transport of 134 Sv estimated by (Cunningham et al., 2003) and 48% of the 40-year mean full-depth model transport of 157 Sv estimated by (Xu et al., 2020).

Transport anomalies for the three components are the largest in the northern half of Drake Passage, especially in the region between the Subantarctic Front and Polar Front (0-500 km, Fig. 4a,d,g). This region has the largest mesoscale eddy activity (meandering of the fronts and eddy formation) within Drake Passage (Watts et al., 2016; Lenn et al., 2011; Foppert et al., 2017; Gutierrez-Villanueva et al., 2020). South of this region the transport anomalies are the weakest (>500 km, Fig. 4a,d,g). While the time series of transport anomalies do not readily show discernible trends, the estimated trends (Fig. 4c,f,i) show clear significant trends consistent with the velocity trends (Fig. 3). All transport components show a clear acceleration of the region between the Subantarctic Front and Polar Front (150-300 km; Fig. 4c,f,i) and deceleration of the southern flank of the Polar Front (350-550 km; Fig. 4c,f,i). Similar patterns are found when total transport trends are calculated with the total velocity transects that coincide with the temperature/salinity transects (Fig. 4c), or using the transects falling along the most repeated crossing (not shown).

Fig. 4 Time series of transport anomaly [Sv], mean transport [Sv] and trends [Sv year⁻¹] per distance bin [km] for October 2005-April 2019. Time series of (a) total, (d) geostrophic, and (g) reference transport anomalies in the upper 760 m. Mean (b) total, (e) geostrophic, and (h) reference transport [Sv] per distance bin. Trends in (c) total, (f) geostrophic, and (i) reference transport [Sv year⁻¹] per distance bin. Red and green lines (b)-(c) show the mean and trends estimated for all (237) available transects and only those (60) velocity transects that have a coincident temperature/salinity transect, respectively. Filled squares (c),(f),(i) indicate that trends are significant at 95% confidence after the modified Mann-Kendall test ($p < 0.05$). Total transport anomalies (a) are estimated based on the mean using all available transects (red line). Standard errors for each line are shown in shaded areas in (b),(e),(h). Mean (\pm standard error) total, geostrophic and reference transport integrated across Drake Passage are indicated with text in (b),(e),(h). Mean Ekman transport (0-90 m) is also indicated in (h). Shaded rectangles show the approximate distance intervals for the Subantarctic Front (SAF), Polar Front (PF), and Southern ACC Front (SACCF) (Gutierrez-Villanueva et al., 2020).

4. Revised Figure 5

The new Fig. 5 is a revision of the original per the suggestion of reviewer 1. The time series show transport anomalies instead of transport. There is a new column showing trends. The content of the original Fig. 6 is now shown in Fig. 5c,f,i.

During the revision, we identified an error in the binning of the transport time series by pair of sea surface height (SSH) streamlines and recalculated the time series, means and trends shown in this figure. The trends estimated using the streamwise coordinates show a much reduced pattern of acceleration/deceleration relative to those estimated in passage coordinates (Fig. 4), with only 2 points showing significance. We use this information, together with the means and trends of eddy kinetic energy (EKE) and eddy momentum flux (EMF), to make a case for eddy saturation state in section 3, after presenting the time series, means and trends of transport in cross-passage coordinates. The modified Fig. 5 and text in section 3 are shown below:

Lines 175-211: The results based on binning the transport by distance indicate that the different ACC frontal regions in Drake Passage have tended to compensate each other over the last 15 years such that the net Drake Passage transport shows no significant acceleration. Nonetheless, the acceleration of the region between the fronts could be an effect of an increase in the meandering of the ACC frontal jets and eddy formation, an aspect not captured when using the fixed cross-passage coordinate system. To explore this hypothesis we constructed time series of transport anomalies and trends analogous to Fig. 4, but where the binning was done per pair of streamlines (Fig. 5), using synoptic sea surface height (SSH) derived from a combination of daily altimetric sea level anomalies and mean geostrophic streamlines (Fig. 1a; see Methods). Contours of SSH provide a natural coordinate system (streamlines) for geostrophic flows. The streamlines allowed us to track the position of the fronts that comprise the ACC, which can vary (both in position and distance between each pair of streamlines) daily due to strong meandering of the fronts and eddy formation (Watts et al., 2016; Gutierrez-Villanueva et al., 2020).

As with the time series of transport anomalies in passage coordinates (Fig.4a,d,g), the transport anomalies in streamwise coordinates (Fig. 5a,d,g) are largest in northern Drake Passage, but are concentrated north of the Polar Front. Similar to the time series in passage coordinates (Fig. 4a,d,g), trends in the streamwise time series are difficult to discern (Fig. 5a,d,g). The magnitude and pattern of the estimated trends for the total transport in streamwise coordinates (Fig. 5c) are much reduced relative to passage coordinates (Fig. 4c), suggesting that at least some of the apparent acceleration in the zone between the Polar and Subantarctic Fronts (Fig. 3, Fig. 4c) was due to fronts meandering and eddies. When binned by streamline, the transport anomalies within meanders and eddies contribute to the anomalies of the fronts rather than the interfrontal zone. The main significant trend in the streamwise coordinate system is that the Subantarctic Front has significantly accelerated over the last 15 years (Fig. 5c). This result is only evident when using the streamwise coordinate system, which is able to capture the effect of the steering of the fronts due to meandering and eddy detachment, an aspect that is missed when using the passage coordinates (Fig. 4).

We will explore the increasing mesoscale activity hypothesis in section 5 by estimating trends in eddy energy and momentum flux. In the following section, we explore buoyancy changes across Drake Passage that may contribute to increased eddy formation.

Fig. 5 Time series of transport anomaly, and mean transport [Sv] per pair of SSH streamlines [m] for October 2005-April 2019. Time series of (a) total, (d) geostrophic, and (g) reference transport anomalies in the upper 760 m. Mean (b) total, (e) geostrophic, and (h) reference transport [Sv]. Trends in (c) total, (f) geostrophic, and (i) reference transport [Sv year⁻¹]. Red and green lines (b)-(c) show the mean and trends estimated for all (237) available transects and only those (60) velocity transects that have a concurrent temperature/salinity transect, respectively. Filled squares (c),(f),(i) indicate that trends are significant at 95% confidence after the modified Mann-Kendall test ($p < 0.05$). Total transport anomalies (a) are estimated based on the mean using all available transects (red line). Standard errors for each line are shown in shaded areas in (b),(e),(h). Mean (\pm standard error) total, geostrophic and reference transport integrated across Drake Passage are indicated with text in (b),(e),(h). Ekman transport is indicated in (h) (see Methods). Shaded rectangles show the streamwise intervals for the Subantarctic Front (SAF), Polar Front (PF), and Southern ACC Front (SACCF) (Gutierrez-Villanueva et al., 2020).

5. New Figure 7

This figure showing depth-averaged EKE and EMF time series, means, and trends is moved from the supplementary information to the main text, as we use it to support our case for an eddy saturation state. The details of the calculation have been moved from the supplementary information to the Methods (8.7 Estimating eddy kinetic energy and eddy momentum flux). The new Fig. 7, the new text in section 3 and methods section are shown below:

Section 5. Trends in eddy activity

Lines 241-282: As discussed in section 3, the acceleration of the transport in the region between the Subantarctic and Polar Fronts and the deceleration of the Polar Front jet (Figs. 3,4) potentially arise from an increase in the steering and meandering of the ACC fronts due to enhanced mesoscale eddy activity. To examine this hypothesis, we constructed time series of depth-averaged eddy kinetic energy and eddy momentum fluxes per pair of streamlines ($\langle \text{EKE} \rangle$ and $\langle \text{EMF} \rangle$, respectively; see supplementary information).

Time series of streamwise $\langle \text{EKE} \rangle$ show that eddy energy increases during the last quarter of the sampling period (Fig. 7a) in the zone between the Subantarctic Front and Polar Front ($-0.60 \text{ m} > \text{SSH} > -1.00 \text{ m}$). This zone exhibits the largest time-mean (denoted by overbar) depth-averaged eddy kinetic energy ($\overline{\langle \text{EKE} \rangle}$), consistent with previous studies (Lenn et al., 2011; Gutierrez-Villanueva et al., 2020; Foppert et al., 2017); $\overline{\langle \text{EKE} \rangle}$ decreases significantly away from this region (Fig. 7b). Trends show a significant increase in $\langle \text{EKE} \rangle$ in the south flank of the Subantarctic Front over the last 15 years (Fig. 7c), indicating an increase in the meandering of the front's streamlines and eddy formation. Surprisingly, trends show a significant decrease of $\langle \text{EKE} \rangle$ in the Polar Front (Fig. 7c). Our means are unaltered when they are calculated using only those (60) velocity transects that have a coincident temperature/salinity transect (Fig. 7b). Trends using the coincident transects show a positive peak centered in the zone between the Subantarctic and Polar Fronts; this peak lies poleward of the significant positive trend estimated using all transects, although trends using the coincident transects are insignificant (Fig. 7c). Our results are consistent with an eddy saturation state, i.e., the net ACC transport remains invariant to the increased wind forcing whereas the eddy activity increases (Hallberg and Gnanadesikan, 2001; Morrison and Hogg, 2013).

Time series of streamwise $\langle \text{EMF} \rangle$ show that the largest negative values occur in the region between the Subantarctic and Polar Fronts, whereas relatively smaller positive values dominate in the Subantarctic Front (Fig. 7d). The time-mean depth-averaged eddy momentum flux $\overline{\langle \text{EMF} \rangle}$ reflects the pattern seen in the time series; it is positive in the Subantarctic Front ($\text{SSH} > -0.65 \text{ m}$) and negative in the region between the fronts ($-0.65 \text{ m} > \text{SSH} > -1.00 \text{ m}$) (Fig. 7e). Within and south of the Polar Front ($\text{SSH} < -1.00 \text{ m}$), $\overline{\langle \text{EMF} \rangle}$ is again positive although its value is close to zero (Fig. 7e). Positive values imply a northward flux of eastward momentum and negative values imply a southward flux of eastward momentum (Lenn et al., 2011). This interpretation suggests that, on average, the eddies in the interfrontal zone redistribute zonal momentum towards the Subantarctic Front and the Polar Front. The positive trend of $\langle \text{EMF} \rangle$ in the southern flank of the Subantarctic Front (Fig. 7f) suggests that eddies have significantly increased the northward flux of zonal momentum in the south flank of the Subantarctic Front over the last 15 years. Similar results are obtained when means and trends are calculated using only those transects that have a coincident temperature/salinity transect (Fig. 7e,f).

Fig. 7 Time series of eddy kinetic energy (EKE) and eddy momentum fluxes (EMF), mean and trends per pair of sea surface height (SSH) streamlines [m] for October 2005-April 2019. Time series of depth-averaged total (a) $\langle \text{EKE} \rangle$ and (d) $\langle \text{EMF} \rangle$ in the upper 760 m [$\text{m}^2 \text{s}^{-2}$]. Mean (b) $\langle \text{EKE} \rangle$ and (e) $\langle \text{EMF} \rangle$ [$\text{m}^2 \text{s}^{-2}$]. Trends in (c) $\langle \text{EKE} \rangle$ and (f) $\langle \text{EMF} \rangle$ [$\text{m}^2 \text{s}^{-2} \text{year}^{-1}$]. Red and green lines (b)-(c) show the mean and trends estimated for all (237) available transects and only those (60) velocity transects that have a concurrent temperature/salinity transect, respectively. Filled squares (c),(f) indicate that trends are significant at 95% confidence after the modified Mann-Kendall test ($p < 0.05$). Standard errors for each line are shown in shaded areas in (b),(e). Shaded rectangles show the streamwise intervals for the Subantarctic Front (SAF), Polar Front (PF), and Southern ACC Front (SACCF) (Gutierrez-Villanueva et al., 2020).

Subsection 8.7 Estimating eddy kinetic energy and eddy momentum flux

Lines 530-547: To quantify trends in the eddy activity in Drake Passage and the eddy momentum transfer with the ACC jets, we estimated time series of depth-averaged eddy kinetic energy and eddy momentum flux from the corrected OS38 ADCP records (see supplementary information eq. 4). We focus on the period of October 2005 - April 2019 (237 velocity transects), which is the period employed to estimate trends in cross-transect velocity and streamwise transport. Following Gutierrez-Villanueva et al. (2020), eddy velocity terms per transect per depth bin were estimated as $\mathbf{u}' = \mathbf{u} - \bar{\mathbf{u}}$, where \mathbf{u} and $\bar{\mathbf{u}}$ are the 5-min velocity vector and the time-mean velocity vector (see Methods in main manuscript). Subsequently, depth-averaged (0-760 m) eddy kinetic energy (EKE) and eddy-momentum flux (EMF) per transect were calculated as

$$\text{EKE} = \frac{1}{760} \int_{-760}^0 0.5 * (u'^2 + v'^2) dz, \quad (6)$$

$$\text{EMF} = \frac{1}{760} \int_{-760}^0 u'v' dz. \quad (7)$$

Next, each EKE and EMF transect was bin-averaged at a ~ 12 km along-track distance and then streamwise averaged using the synoptic SSH to construct time series of $\langle \text{EKE} \rangle$ and $\langle \text{EMF} \rangle$ per transect per pair of streamlines, where the angle brackets $\langle \cdot \rangle$ denote the bin and streamwise averaging. The ~ 12 km resolution allows us to meet the criterion of at least one point per pair of streamlines per transect.

6. New Figure 8

We have included a schematic summarizing the response of the Antarctic Circumpolar Current (ACC) to changes within the frontal regions over the last 15 years as discussed in the paper. The new Fig. 8 and the modified text in the discussion are shown below:

Lines 303-317: Trends of depth-averaged EKE estimated from the velocity transects show that eddy activity has increased over the last 15 years in the zone between the Subantarctic and Polar Fronts, peaking on the southern flank of the Subantarctic Front (Fig. 7c). The increased EKE is consistent with altimetry-based EKE trends observed in ACC eddy hot spots like Drake Passage over the last two decades (Martínez-Moreno et al., 2021; Hogg et al., 2015). Our interpretation is consistent with an eddy saturation state (i.e., eddy activity increases as the wind stress increases with no net acceleration of the ACC; Hallberg and Gnanadesikan, 2001; Meredith and Hogg, 2006; Patara et al., 2016; Meredith et al., 2012; Downes and Hogg, 2013; Munday et al., 2013; Morrison and Hogg, 2013) and with observed buoyancy changes (Fig. 6). The warming in the Subantarctic Front (red patch, Fig. 8a) and wind-driven cooling south of the Polar Front (blue patch, Fig. 8b) act to steepen the isopycnals across the fronts (dashed tilted, Fig. 8a,b), enhancing the available potential energy stored in the fronts that is subsequently released through baroclinic instabilities (Vallis, 2017). Consequently, more baroclinic instabilities translate into more steering and meandering of the fronts and eddy formation (Fig. 8c), which feed into the vigorous eddy field in Drake Passage (Gutierrez-Villanueva et al., 2020; Watts et al., 2016; Foppert et al., 2017; Lenn et al., 2011).

Fig. 8 The response of the Antarctic Circumpolar Current to buoyancy changes within the ACC frontal regions over the last 15-years. Buoyancy changes across the (a) Subantarctic Front (SAF) and (b) Polar Front (PF). Red patch in (a) shows warming and blue patch in (b) show wind-driven cooling. Blue thick horizontal arrow in (b) is the intensified equatorward Ekman transport due to increasing westerly wind stress τ_x (circles). Blue vertical arrows in (b) show the enhanced wind-driven upwelling and shoaling of isopycnals. Solid black lines are the fronts isopycnals γ ($\gamma_1 < \gamma_2 < \gamma_3$) in normal conditions; dashed lines are steepened isopycnals due to the buoyancy changes in the fronts. Surface thick solid lines are the sea surface height (SSH) in normal conditions; thick dashed lines are the SSH of the steepened fronts' SSH. (c) Eddy saturation state in Drake Passage. Solid gold and gray contours show the SAF and PF SSH streamlines, respectively, with arrow heads showing the flow direction within the frontal jets. Filled gold and silver vortices are mesoscale eddies detached from the SAF and PF, respectively.

7. Ekman transport

We have included an estimate of the Ekman transport in Figs. 4h and 5h, such that the Ekman, geostrophic, and reference transports sum to the total. We have modified the Methods (Subsection 8.4) to include the Ekman transport calculation. The new methods subsection reads:

Subsection 8.4 Total, geostrophic and reference transport

Lines 452-469: The total cross-transect velocity u_{tot} , defined as that of the ADCP velocities, can be decomposed into geostrophic u_{geo} , reference u_{ref} and Ekman u_{Ekm} components (i.e. $u_{tot} = u_{geo} + u_{ref} + u_{Ekm}$) when XBT/XCTD transects are available. We estimated the Ekman component by subtracting the geostrophic component profile from the total component profile (i.e., $u_{Ekm} = u_{tot} - u_{geo}$; Lenn and Chereskin, 2009) within the Ekman layer ($z \leq z_{Ekm}$). The Ekman layer was defined where the total shear showed an exponentially decaying profile, after first averaging per transect and then averaging for the entire time series (Lenn and Chereskin, 2009). For our observations, the base of the Ekman layer was $z_{Ekm} = -90$ m since the mean shear velocity reduces to a constant value below 90 m and is in good agreement with the geostrophic shear. Here, we interpolated the total velocity to the geostrophic velocity along-track and depth grid for each transect. Thus the Ekman transport was calculated as $U_{Ekm} = \int_0^L \int_{z_{Ekm}}^0 u_{Ekm} dz dx$. Consequently, the reference component u_{ref} was defined as the averaged residual velocity between the base of the Ekman layer and the deepest bin z_0 :

$$u_{ref} = \frac{1}{|z_{Ekm} - z_0|} \int_{z_0}^{z_{Ekm}} (u_{tot} - u_{geo}) dz. \quad (3)$$

Therefore, the reference Drake Passage transport is $U_{ref} = \int_0^L \int_{z_0}^{z_{Ekm}} u_{ref} dz dx$.

Comments by Reviewer 1

In “Compensating transport trends in the Drake Passage frontal regions yield no acceleration in net transport”, Gutierrez-Villanueva et al. presented the Drake Passage transport calculated from 15 years of direct measurements (2005-2019) for the top 760 m, based on shipboard ADCP surveys along with XBT/XCTD measurements (that covered from 1997 to 2019). The key finding is that while the net transport measured exhibits no acceleration, the key front regions do exhibit statistically significant trends that have opposite signs. They further suggested that the acceleration of the frontal jets results from an increase in the mesoscale eddy activity due to buoyancy changes in the fronts, consistent with an eddy saturation state. I believe this observational work is of great value to the community and, to some degree, I think the authors can emphasize a bit more on the strength and/or the importance of this unique observation, i.e., high-resolution direct measurement of the current/transport across the full passage (for the upper ocean). However, there are some places I think the point is a bit unclear or potentially misleading, hence some clarifications are needed.

General comments

1: The compensating trends in frontal regions “explain” (L20) or “are likely responsible for” (L69) the lack of trend in the net transport. Probably a minor wording issue but I am not sure if this causative implication is appropriate. The net transport is made of the transports of the front regions, so it is somewhat strange to call one explains/is responsible for the other. Simply stating the results, i.e., no trends in the net transport, but compensating trends in the frontal regions, is more appropriate to me.

Reply: We agree with the reviewer and have modified the abstract to:

Lines 17-20: We found that, although the net Drake Passage transport relative to 760 m shows insignificant acceleration, the net transport trend comprises compensating trends across the ACC frontal regions.

We also modified our discussion to be consistent with the abstract, and added a new figure (Fig. 8) which shows a schematic of the response of the ACC in Drake Passage to the buoyancy changes over the last 15 years. We refer the reviewer to the Major changes section (6. New Figure 8) in this response to read the modified text and new Fig. 8 (Pages 13, Lines 303-317).

2: I do not think it is accurate to state “SAF and PF jets have accelerated while the currents located between the front have decelerated . . .” (L17-18). If one reads Fig. 3 correctly, the strongest increase is located between SAF and PF, the southern half of the PF and the region between PF and SACCF show decreasing velocity (L130-134; L185-186).

Reply: We thank the reviewer for raising this point. The confusion stems because of the projection of the trends in (now) Fig. 4 (see response to the next comment) in distance across the passage [km], which does not account for the meandering of the ACC front’s streamlines and eddy formation. We refer the reviewer to the Major changes section (3. New Figure 4) in this response to read the modified text in section 3 (Pages 6-7, Lines 146-174). Also, we have moved Fig. 5 into section 3 and discussed the acceleration of the region

between the fronts found in Fig. 3, 4 as an effect of an increase in the meandering of the ACC frontal jets and eddy formation. We refer the reviewer to the Major changes section (4. Revised Figure 5) to read the modified text (Pages 8-9, Lines 175-211).

3: Not sure if “acceleration of frontal jets results from an increase of mesoscale eddy activity” is consistent with eddy saturation state (L22-23). If an increase of mesoscale eddy activity leads to acceleration of the frontal jets, it should ultimately lead to a stronger net ACC transport, right? And what contributes the decreasing trend in the southern part of PF and north of SACCF?

Reply: We have modified our discussion about the interpretation of the eddy momentum flux across the frontal regions. We refer the reviewer to the Major changes section in this response (5. New Figure 7) to read the modified text (Page 10, Lines 266-282), where we have addressed the reviewer’s comment. Likewise, we have modified the section 6 where we discuss the trends in EMF. The modified text reads:

Lines 318-329: Moreover, the mean pattern of eddy momentum flux shows that eddies have acted to redistribute momentum across the ACC frontal regions over the last 15 years, by removing momentum from the interfrontal zone and depositing it in the jets (Fig. 7e). The positive trend in eddy momentum flux at the southern boundary of the Subantarctic Front could help to accelerate the jet (Fig. 7f). The role of eddy momentum flux in redistributing momentum due to increased winds and its effect on the ACC jets is an outcome not covered in the eddy saturation state hypothesis and warrants future work. Eddy-permitting models with sufficient resolution to resolve the ACC fronts in Drake Passage (Xu et al., 2020), forced with both realistic and idealized wind-forcing, provide an opportunity to study the potential eddy-driven sharpening of the ACC fronts posed in this study.

4: I wonder if it helps to remove the mean from Fig 5, because it is difficult to see trend (which might be the reason to have Fig. 6). More importantly, I think it is valuable to have a similar time series plot from the geophysical coordinate perspective, showing over time the change of vertically integrated velocity that corresponds to the trend in Fig. 3 (This plot can even include those sections that did not cross 1000 m isobath if those sections have been processed, just not included in transport calculation).

Reply: We thank the reviewer for the suggestions. We refer the reviewer to the Major changes section (4. Revised Figure 5) to read the modified text (Pages 8-9, Lines 175-211) where we have addressed the reviewer’s comment.

5: The ADCP based transport is calculated for the top 760 m, not to 1000 m as in the previous work covering 4.5 years of the observations (Firing et al., 2011). I assume this is because many sections do not extend to 1000 m, otherwise, one may argue it is valuable to have a separate time series that cover the top 1000 m to be consistent with the previous work (separating between geostrophic and reference transport at 760 m seems of less importance to me, although that is arguable).

Reply: We could not reliably extend our calculations to 1042 m given the large gaps in velocity data below 970 m. We have provided some clarification for the criterion used to discard velocity transects. The modified text reads:

Lines 350-361: Transects were discarded if one of the following criteria were met: a) transects were not completed in less than 4 days, b) along-track/vertical gaps were too large to fill, or c) if transects with gaps outside the area enclosed by the gridded objectively mapped mean (see subsection 8.3). Following these criteria allows us to estimate mean and trends in both the net total transport and transport per distance bin with the same number of degrees of freedom (transects). From February 2005 - December 2019, 248 (of 286 total crossings) ADCP transects met these criteria, with 147 transects along the most commonly repeated line (Supplementary Fig. S1). From October 2005 - April 2019, the common sampling period between the ADCP and the XBT/XCTD transects (details below), 237 transects were available with 140 transect falling along the most repeated line.

We found that both the transport bias and corrected mean net Drake Passage total transport are in close agreement with that of Firing et al. (2011) (SI Lines 42-45). Because we modified Section 3 and added Fig. 4, we moved the discussion about extending the transport calculation to 970 m previously in Section 5 to section 3. We refer the reviewer to Major changes section (3. New Figure 4) in this response where we discussed the mean transport in the upper 970 m (Page 6, Lines 152-157).

We focused on the upper 760 m since it is the maximum common depth between the XBT/XCTD and cross-transect total velocity profiles (Lines 401-404).

Other details

1: L39. *Ref 4 may be placed at the end of this sentence as it is about both ozone and greenhouse gas emissions (It should be mentioned that it is a modeling work). Not sure why ref 5 (also a modeling work) is included here as it is mostly about ocean warming, not so much about wind or southern annular mode.*

Reply: We have removed ref. 5 as suggested and modified the sentence:

Lines 41-43: Modeling studies show that the positive trend in the winds is a response to both depletion of the atmospheric ozone and increments in the greenhouse gas emissions over the last decades (Sigmond et al., 2011; Saenko et al., 2005; Gillett and Thompson, 2003).

2: L58, *“... time series were not yet long enough” and L111. “... using short sampling periods” are not accurate for refs. 16 and 19. The geostrophic transports based on hydrographic surveys (ref 16) cover a pretty long period (although less frequent). Ref 19 rely much on satellite SSH which is not direct measurement of current. I think the strength or value of this ‘unique’ observation needs to be highlighted a bit more, i.e., high-resolution, direct measurements of the current across the full Drake passage over a long period of time (One limitation is that it covers only upper part of the water column).*

Reply: We thank the reviewer for pointing out this mistake. We have modified the manuscript such that it reflects the value of the Drake Passage time series.

Lines 59-67: However, to date, none of the observational studies have found a clear trend in the net Drake Passage transport (Cunningham et al., 2003; Chidichimo et al., 2014; Donohue et al., 2016; Koenig et al., 2014; Meredith and Hogg, 2006). Moreover, these studies were unable to explore trends within the ACC frontal regions owing to the lack of direct current velocity measurements at a sufficient spatial resolution to resolve the frontal regions and mesoscale activity. The lack of long-term in-situ observations (velocity, temperature, salinity) with high horizontal resolution in the SO hampers the exploration of trends in the ACC transport and properties across the different frontal regions.

Similarly, we modified the text within section 2 that now reads:

Lines 119-124: Our results are consistent with previous Drake Passage observational studies using coarser resolution in-situ observations (Cunningham et al., 2003; Chidichimo et al., 2014) and altimetry-derived geostrophic currents (Koenig et al., 2014) which found no significant acceleration of the Drake Passage transport.

We also modified section 3 in the manuscript to clarify that our results are representative of the upper 760 m:

Lines 68-73: This study uses a unique observational time series of year-round near-repeat upper-ocean temperature, salinity, and velocity across the Drake Passage (Fig. 1a) to explore whether the Drake Passage transport in the upper 760 m shows significant acceleration in the last 15 years. The repeated upper-ocean velocity transects provide the most efficient way to measure the upper-ocean transport (Xu et al., 2020).

3: L113. Refs. 21, 22 find no acceleration in the Drake passage but their main result is that the warming is leading to acceleration in zonal flow in much of the Southern Ocean, and I do not think low-resolution climate model is the best examples to show no acceleration in ACC transport (because without eddies, a stronger wind tends to drive a stronger ACC transport).

Reply: We agree with the reviewer that coarse-resolution global climate models show no acceleration because they are unable to resolve mesoscale activity. Therefore, we removed lines 113-114 from the original manuscript to avoid confusion.

4: L114-L120. Ref. 23 found that “baroclinic and barotropic transports are not correlated; thus, monitoring either baroclinic or barotropic transport alone may be insufficient to assess the temporal variability of the total ACC transport”. They did not imply that the lack of trend in their modeled transport is due to two components compensating each other. Furthermore, the separation of geostrophic and reference velocity at 760 m has different meaning from the baroclinic - barotropic separation for full water column. On a different perspective, I think it is noteworthy that ref 23 carefully evaluated the modeled ACC transport structure using various observations, including the result of the first 4.5 years of the ADCP measurement that are presented in this paper. The agreement between model and the ADCP measurement for the top 1000 m was one key reason that gives the confidence in their modeled transport. Thus, the observations like this are also of great value to modeling community as well. I think this point could be noted somewhere in the paper.

Reply: We agree with the reviewer. We removed Lines 113-120 from the manuscript. Regarding the comparisons between the modeled ACC and the ADCP observations from (Firing et al., 2011) by Xu et al. (2020), we moved this discussion into section 3 since we added Fig. 4 and included a few lines that address the reviewer's suggestion. We refer the reviewer to the Major changes section (3. New Figure 4) where we have addressed the reviewer's comment in section 3 (Page 6, Lines 152-157). Likewise, we added text in the discussion that addresses the reviewer's comment. The added text reads:

Lines 325-329: Eddy-permitting models with sufficient resolution to resolve the ACC fronts in Drake Passage (Xu et al., 2020), forced with both realistic and idealized wind-forcing, provide an opportunity to study the potential eddy-driven sharpening of the ACC fronts posed in this study.

5: L176. “[32] found a similar . . .” you may rephrase this as “a similar cooling . . . [32]”.

Reply: We modified the sentence per the reviewer's suggestion. Now it reads:

Lines 231-233: A similar cooling trend in sea surface temperature in Drake Passage was found from 1993 to 2017 (Auger et al., 2021).

6: Fig.1 I assume panels b-d is along the common section? If yes, it should be mentioned in the Figure caption and noted that the location of this section is indicated in S1, or add a one line in panel a.

Reply: The climatological means were calculated using all available total velocity, temperature/salinity transects, i.e. including also those transects away from the most repeated line, for the periods mentioned in the figure caption. Likewise, all time series and trends are calculated by employing all transects (see Fig. 1's caption).

Comments by Reviewer 2

This paper uses a 15-year time series of subsurface temperature, salinity, and velocity in the Drake passage to investigate the structure of the transport trends of the Antarctic Circumpolar Current. The paper presents the trends in the overall net transport across the Drake Passage, cross-passage velocity and streamwise transport, and link them with the buoyancy changes across the passage.

These results hold significant importance as they offer a detailed description of the transport trends in the Drake Passage, particularly in relation to the ACC's response to strengthening winds, which remains an active research question. It is worth noting that although the Drake Passage is the most regularly sampled section of the Southern Ocean, long-term time series of temperature, salinity, and velocity across the entire Southern Ocean are scarce and highly valuable. The paper is well written and only a few minor issues need to be addressed or clarified in my opinion.

Comments/Questions

1: *The paper frequently refers to the impact of wind on buoyancy and velocity change. Have the authors attempted to compute trends in winds or wind stress curl along the transect? If so, how consistent are these trends with Figure 4 and the changes described in Section 4?*

Reply: We thank the reviewer for suggesting estimating trends in wind stress or wind stress curl. We did not include them in the main manuscript or supplementary information since it is well documented that the wind stress over the Southern Ocean (as shown by the Southern Annular Mode, SAM, the largest empirical mode of westerly wind stress in the region) has increased (Marshall, 2003; Fogt and Marshall, 2020) for the last four decades which cover our period of study. However we include the trends in the wind stress curl as they are important when calculating trends in spice and heave components of temperature and salinity in Drake Passage (see Fig S4).

Fig. S2 Wind stress curl over the Southern Ocean for 2005-2019. (a) Time-mean wind stress curl $\mathbf{k} \cdot \nabla \times \boldsymbol{\tau}$ [N m^{-3}]. Negative values indicate Ekman suction (upwelling). (b) Trend in wind stress curl [$\text{N m}^{-3} \text{ year}^{-1}$] for the 2005-2019. Negative values indicate increasing Ekman suction. Trends enclosed with gray solid contour are 95% statistically significant after the modified Mann-Kendall test.

We also modified section 4 to include the results shown in the supplementary figure in the manuscript. The modified text now reads:

Lines 228-231: The heave-driven cooling potentially stems from a more negative wind-stress curl (Ekman suction or upwelling) over the Drake Passage area during the last 15 years (Supplementary Fig. S2), consistent with previous trends estimated for a shorter period of time (Meehl et al., 2019).

2: L144: “is located”?

Reply: In response to a suggestion from Reviewer 1, we have added a new figure in section 3 (Fig. 4) and its subsequent description in Section 3 (see our following response). Hence, this typo has been deleted.

3: Figure 3 and Section 3 may confuse readers. While Figure 3 sets the methodology for separating the total, reference, and geostrophic components, and shows in a simple way that the compensation of the ACC frontal regions, it presents results that, at first glance, seem to contradict the final conclusions. Specifically, it shows a deceleration of velocities in the SAF and PF zones, as well as an acceleration in the region between them, whereas the abstract mentions that the subantarctic and polar jets have accelerated while the currents between them have decelerated. While this might be confusing at first read, this confusion could be partly avoided by adding a few similar words as the opening of section 5. I leave this decision up to the authors.

Reply: We thank the reviewer for raising this matter. In response to a similar comment raised by reviewer 1, we have moved Fig. 5 into section 3 and discussed the acceleration of the region between the fronts found

in Fig. 3, 4 as an effect of an increase in the meandering of the ACC frontal jets and eddy formation. We refer the reviewer to the Major changes section (4. Revised Figure 5) in this response to read the modified text (Pages 8-9, Lines 175-190) where we have addressed the reviewer's comment.

4: L173-175: *The difference in magnitude between Fig. 4 and Fig. S2 is surprising. Does that question the robustness of the trends?*

Reply: We respectfully disagree with the reviewer. While the difference in magnitude between Fig. 4 and Fig. S2 is stark, statistically significant trends shown in the figures are similar both in sign and spatial distribution. This gives us confidence that the long-term trends are robust. Nevertheless, we have decided to remove Fig. S2 from the supporting information and lines 173-176 in the text to avoid any confusion.

5: L283: *“to rise”*

Reply: We have corrected this typo. Now the sentence reads:

Lines 335-339: Given the sustained increase in global greenhouse emissions, the westerly wind stress over the ACC is expected to continue to rise, which would act to increase the poleward eddy transport and thus the upwelling of deep warm waters along isopycnals outcropping near the Antarctic continental shelf (Palóczy et al., 2018; Tamsitt et al., 2016).

6: L299: *Is there any specific criterion used to discard the transects that significantly deviated from the main course?*

Reply: We have modified the Methods section to better reflect the specific criteria used to discard transects. The modified text reads:

Lines 350-361: Transects were discarded if one of the following criteria were met: a) transects were not completed in less than 4 days, b) along-track/vertical gaps were too large to fill, or c) if transects with gaps outside the area enclosed by the gridded objectively mapped mean (see subsection 8.3). Following these criteria allows us to estimate mean and trends in both the net total transport and transport per distance bin with the same number of degrees of freedom (transects). From February 2005 - December 2019, 248 (of 286 total crossings) ADCP transects met these criteria, with 147 transects along the most commonly repeated line (Supplementary Fig. S1). From October 2005 - April 2019, the common sampling period between the ADCP and the XBT/XCTD transects (details below), 237 transects were available with 140 transect falling along the most repeated line.

7: L338-340: *How many of the transects between October 2005 and December 2019 fall into the most sampled line? I am just wondering how the zonal spreading of the transects could impact the cross-transect velocity, temperature and salinity trends. More specifically, can that explain part of the differences in the trends from the longer and shorter time periods (Fig. 3a vs Fig Fig.3b; Fig.4 vs Fig S2).*

Reply: Between October 2005 - April 2019, only 26 (45%) of the 69 temperature/salinity transects and 140 (60%) of the 237 and ADCP transects fall along the most repeated line. We did not calculate trends for

temperature/salinity for the most repeated line due to the small percentage of transects and gaps in the time series of as much as 2 years. In response to the reviewer’s comment, we estimated trends in the total transport per distance bin using the transects falling along the most repeated line and they were similar in spatial distribution, magnitude, and sign compared to those calculated using all transects available (Fig. R1). We conclude that the zonal spreading of the transects does not impact significantly the total transport trends found in this study.

Fig. R1 Trends in cross-transect total transport per distance bin [Sv year^{-1}] from October 2005 - April 2019. Red and green lines are trends estimated using all available (237) velocity transects and only those (140) velocity transects that fall along the most repeated line. Shaded areas show the error bars estimated from the least-squares fit. Filled squares and dots indicate that trends are significant at 95% confidence after the modified Mann-Kendall test ($p < 0.05$). The mean location of the Subantarctic Front (SAF), Polar Front (PF), and Southern ACC Front (SACCF) are indicated by shaded areas.

We modified section 3 in the main manuscript to make our results clear:

Lines 171-174: Similar patterns are found when total transport trends are calculated with the total velocity transects that coincide with the temperature/salinity transects (Fig. 4c), or using the transects falling along the most repeated crossing (not shown).

8: L425-428: *What is the motivation behind the use of two different datasets sources for the Mean Dynamic Topography and Sea Level Anomaly? This could introduce discrepancies between the datasets that could be avoided with the use of the full SSH signal from SSALTO/DUACS.*

Reply: We use the mean dynamic topography because it is directly tied to the ocean’s surface geostrophic circulation. The full Sea Surface Height signal from SSALTO/DUACS is referenced to the ellipsoid and contains the geoid signal which could introduce discrepancies in defining anomalies such as mesoscale meanders and eddies, therefore affecting our binning and attribution methodology. The mean dynamic topography removes the geoid model such that the mean topography is defined exclusively by the mean major surface currents. We have modified subsection 8.6 to justify our choice of using daily maps of sea level anomalies plus the mean dynamic topography. The text now reads:

Lines 491-509: To determine the trends across Drake Passage as a function of the ACC frontal regions, we employed a synoptic streamwise coordinate system (sea surface height; SSH) in a similar way as used by Gutierrez-Villanueva et al. (2020) for eddy heat flux. We used a combination of the mean dynamic topography (Maximenko et al., 2009) and the SSALTO/DUACS daily maps of sea level anomalies from AVISO to track the position of the ACC streamlines. The mean dynamic topography is directly tied to the large-scale surface geostrophic circulation constrained by 20 years of in-situ observations such as drifters corrected due to wind/Ekman currents, altimetry, Argo temperature/salinity profiles, and conductivity-temperature-depth (CTD) casts (Maximenko et al., 2009). The daily maps were obtained from multiple satellite altimeters and objectively mapped to a $0.25^\circ \times 0.25^\circ$ Cartesian grid (Ducet et al., 2000). The sea level anomalies are relative to a twenty-year mean of the sea surface height field. Adding the mean dynamic topography to the sea level anomalies produces maps of dynamical SSH that enable tracking of the shifting and meandering of the ACC streamlines and mesoscale eddies. We used the daily maps from February 1999 to December 2019, which covers our period of interest, and we subtracted a 0.6 m constant from the daily maps following Gutierrez-Villanueva et al. (2020) (and references therein), which does not affect the position of the fronts and their horizontal gradients.

9: L447-449: Does that mean that profiles falling into closed contours are discarded?

Reply: No, the method developed in Gutierrez-Villanueva et al. (2020) bins the profiles falling within closed contours (eddies) into the pair of open contours enclosing the closed contours. We refer the reviewer to Appendix A in Gutierrez-Villanueva et al. (2020) where the binning of data profiles is detailed with examples.

10: L459: You might mean a_0 instead of a_1 .

Reply: We thank the reviewer for pointing out this typo. We replaced a_0 for a_1 so it is consistent with our methodology. The modified text reads:

Lines 552-554: The trends were estimated by least-squares fitting $\hat{y} = a_0 + a_1 t$ to each time series; the second term a_1 on the right-hand side of the equation is the trend.

11: Figure 3. caption: “Trends in (a) and (b) are estimated using all available (237) velocity transects and only those (60) velocity transects that have a coincident temperature/salinity transect used to compute the geostrophic u_{geo} component in (c), respectively.” Please split this sentence into two parts: Trends in (a)... while trends in (b)...

Reply: We thank the reviewer for the suggestion. We wish to refer the reviewer to Major changes section (2. Revised Figure 3) to read the corresponding changes to Figs. 3 where we have addressed the reviewer’s comment (Page 5).

Comments by Reviewer 3

This paper analyses a 15-year time series of Drake Passage made up of underway ADCP measurements and XBT/XCTD deployments. The data has been collected during crossings of ARSV Laurence M. Gould during transits from South America to the Antarctic Peninsula. The measurements cover the top 1000 m, or so, of the water column with analysis restricted to between depths of 20-40 m and 760 m, due to instrument constraints and quality control processing. The resulting transects clearly show the 3 main jets of the Antarctic Circumpolar Current (ACC) and the steeply inclined isopycnals that accompany them. Analysis of least square trends shows that there are no meaningful trends in the Drake Passage transport, consistent with previous publications. However, there are statistically significant trends in the cross-transect velocity, which compensate when the area-integrated cross-transect transect is calculated. The locations of these trends indicates a statistically significant acceleration of the Polar Front and Subantarctic Front, with compensating deceleration between the fronts. This is interpreted as being a result of increased wind stress leading to locally steepened isopycnals and increased Eddy Kinetic Energy (EKE), with the more vigorous eddy field then sharpening and accelerating the jets due to their momentum transport.

My expertise is not in analysing and processing observational data. This makes it difficult for me to comment on these aspects of the paper. The methods section appears thorough and well thought out. It is supported by citations to previous applications of similar techniques with similar corrections, etc, being found necessary in the current paper. The statistical tests are consistently applied and systematically reported. With my inexperienced eye, this all seems quite reasonable.

The paper makes a valuable contribution to understanding how the Southern Ocean and ACC has and/or will respond to climate change. Their results are consistent with previous ones showing no increase in ACC transport. However, they are able to add the important detail that this may be because of compensation between different regions increasing/decreasing their velocity. The authors link this to eddy-driving of the jets via eddy momentum fluxes and provide quantification of the same.

General comments

1: There are some points missing from the discussion that I think should be included. The observations only cover the top 760 m of the water column, amounting to 61% of the baroclinic transport or 47% of the transport including the near-bottom flow (lines 202-204). However, there is no discussion of whether the results are likely to hold over the whole water column. Is it likely that the same pattern of spatially localised accelerations and decelerations hold?

Reply: In response to a similar comment raised by reviewer 2, we have provided some clarification of our results as to how representative they are for the whole water column. We refer the reviewer to Major changes section (3. Revised Fig. 3) in this response to reading the modified text (Pages 4, Lines 132-134).

2: The second thing that might warrant some discussion is whether the conclusions would hold upstream/downstream of Drake Passage. A recent paper, Shi et al. (2021), indicates that there might be acceleration of the northern edge of the ACC elsewhere in the Southern Ocean. Can the authors results shed any light on whether this might affect the circumpolar transport of the ACC? Or is this

too speculative, given limited Southern Ocean observations?

Reply: We do not think we can tie the results by Shi et al. (2021) with ours because they employed a coarse-resolution global climate model that are incapable of resolving eddies, and therefore unable to resolve any eddy-driven acceleration in Drake Passage and the ACC as detected in our high-resolution observations.

Minor Comments

There were a few minor things I noticed whilst reading the paper that the authors may wish to consider.

1: Lines 73-76: This sentence seems incomplete, did the authors mean to point towards Figure 1?

Reply: We thank the reviewer for pointing this out. The modified sentence reads:

Lines 82-85: Climatological mean sections of cross-transect velocity u_{tot} , potential temperature θ (temperature hereinafter), and salinity S across Drake Passage using all available velocity (237) and temperature/salinity (114) transects (see Methods) are shown in Fig. 1(b-d).

2: Figure 5b: I found it hard to tell the lines apart, because their colour is the same, even with one having prominent markers. Different colours might be better.

Reply: We thank the reviewer for pointing us to this issue. In response to comments by reviewers 1 and 2, we have moved Fig. 5 into section 5. We wish to refer the reviewer to Major changes section (4. Revised Figure 5) to see the revised Fig. 5 where we have addressed the reviewer's comment (Page 9).

3: Lines 148-151 and Figure 3b: I was puzzled as to the difference between Figure 3a and 3b until I got to this point. It might be better to put this information in the figure caption or move it earlier in the paragraph, since readers may look at the figure before reading the full paragraph (as I did).

Reply: We thank the reviewer for the suggestion. We wish to refer the reviewer to the Major changes section (2. Revised Figure 3) to see the changes to Fig. 3 (Page 5).

References

- Auger, M., Morrow, R., Kestenare, E., Sallée, J.-B., and Cowley, R. (2021). Southern Ocean in-situ temperature trends over 25 years emerge from interannual variability. *Nat. Commun.*, 12(1):1–9.
- Chidichimo, M. P., Donohue, K. A., Watts, D. R., and Tracey, K. L. (2014). Baroclinic transport time series of the Antarctic Circumpolar Current measured in Drake Passage. *J. Phys. Oceanogr.*, 44(7):1829–1853.
- Cunningham, S. A., Alderson, S. G., King, B. A., and Brandon, M. A. (2003). Transport and variability of the Antarctic Circumpolar Current in Drake Passage. *J. Geophys. Res. Oceans*, 108(C5).
- Donohue, K. A., Tracey, K. L., Watts, D. R., Chidichimo, M. P., and Chereskin, T. K. (2016). Mean Antarctic Circumpolar Current transport measured in Drake Passage. *Geophys. Res. Lett.*, 43(22):11,760–11,767.
- Downes, S. M. and Hogg, A. M. (2013). Southern Ocean Circulation and Eddy Compensation in CMIP5 Models. *J. Clim.*, 26(18):7198 – 7220.
- Ducet, N., Le Traon, P.-Y., and Reverdin, G. (2000). Global high-resolution mapping of ocean circulation from TOPEX/Poseidon and ERS-1 and-2. *J. Geophys. Res.*, 105(C8):19477–19498.
- Firing, Y. L., Chereskin, T. K., and Mazloff, M. R. (2011). Vertical structure and transport of the Antarctic Circumpolar Current in Drake Passage from direct velocity observations. *J. Geophys. Res.*, 116(C8).
- Fogt, R. L. and Marshall, G. J. (2020). The Southern Annular Mode: Variability, trends, and climate impacts across the Southern Hemisphere. *WIREs Clim. Chg.*, 11(4):e652.
- Foppert, A., Donohue, K. A., Watts, D. R., and Tracey, K. L. (2017). Eddy heat flux across the Antarctic Circumpolar Current estimated from sea surface height standard deviation. *J. Geophys. Res. Oceans*, 122(8):6947–6964.
- Gillett, N. P. and Thompson, D. W. (2003). Simulation of recent Southern Hemisphere climate change. *Science*, 302(5643):273–275.
- Gutierrez-Villanueva, M. O., Chereskin, T. K., and Sprintall, J. (2020). Upper-ocean eddy heat flux across the Antarctic Circumpolar Current in Drake Passage from observations: time-mean and seasonal variability. *J. Phys. Oceanogr.*, 50(9):2507–2527.
- Hallberg, R. and Gnanadesikan, A. (2001). An Exploration of the Role of Transient Eddies in Determining the Transport of a Zonally Reentrant Current. *J. Phys. Oceanogr.*, 31(11):3312 – 3330.
- Hogg, A. M., Meredith, M. P., Chambers, D. P., Abrahamsen, E. P., Hughes, C. W., and Morrison, A. K. (2015). Recent trends in the southern ocean eddy field. *J. Geophys. Res. Oceans*, 120(1):257–267.
- Killworth, P. D. and Hughes, C. W. (2002). The Antarctic Circumpolar Current as a free equivalent-barotropic jet. *Journal of marine research*, 60(1):19–45.
- Koenig, Z., Provost, C., Ferrari, R., Sennéchaël, N., and Rio, M.-H. (2014). Volume transport of the Antarctic Circumpolar Current: Production and validation of a 20 year long time series obtained from in situ and satellite observations. *J. Geophys. Res. Oceans*, 119(8):5407–5433.

- Lenn, Y.-D. and Chereskin, T. K. (2009). Observations of Ekman Currents in the Southern Ocean. *J. Phys. Oceanogr.*, 39(3):768 – 779.
- Lenn, Y.-D., Chereskin, T. K., Sprintall, J., and McClean, J. L. (2011). Near-surface eddy heat and momentum fluxes in the Antarctic Circumpolar Current in Drake Passage. *J. Phys. Oceanogr.*, 41(7):1385–1407.
- Marshall, G. J. (2003). Trends in the Southern Annular Mode from Observations and Reanalyses. *J. Clim.*, 16(24):4134 – 4143.
- Martínez-Moreno, J., Hogg, A. M., England, M. H., Constantinou, N. C., Kiss, A. E., and Morrison, A. K. (2021). Global changes in oceanic mesoscale currents over the satellite altimetry record. *Nat. Clim. Chang.*, 11(5):397–403.
- Maximenko, N., Niiler, P., Centurioni, L., Rio, M.-H., Melnichenko, O., Chambers, D., Zlotnicki, V., and Galperin, B. (2009). Mean dynamic topography of the ocean derived from satellite and drifting buoy data using three different techniques. *J. Atmos. Oceanic Technol.*, 26(9):1910–1919.
- Meehl, G. A., Arblaster, J. M., Chung, C. T., Holland, M. M., DuVivier, A., Thompson, L., Yang, D., and Bitz, C. M. (2019). Sustained ocean changes contributed to sudden Antarctic sea ice retreat in late 2016. *Nat. Commun.*, 10(1):1–9.
- Meredith, M. P., Garabato, A. C. N., Hogg, A. M., and Farneti, R. (2012). Sensitivity of the Overturning Circulation in the Southern Ocean to Decadal Changes in Wind Forcing. *J. Clim.*, 25(1):99 – 110.
- Meredith, M. P. and Hogg, A. M. (2006). Circumpolar response of Southern Ocean eddy activity to a change in the Southern Annular Mode. *Geophys. Res. Lett.*, 33(16):L16608.
- Morrison, A. K. and Hogg, A. M. (2013). On the Relationship between Southern Ocean Overturning and ACC Transport. *J. Phys. Oceanogr.*, 43(1):140 – 148.
- Munday, D. R., Johnson, H. L., and Marshall, D. P. (2013). Eddy Saturation of Equilibrated Circumpolar Currents. *J. Phys. Oceanogr.*, 43(3):507 – 532.
- Palóczy, A., Gille, S. T., and McClean, J. L. (2018). Oceanic Heat Delivery to the Antarctic Continental Shelf: Large-Scale, Low-Frequency Variability. *J. Geophys. Res. Oceans*, 123(11):7678–7701.
- Patara, L., Böning, C. W., and Biastoch, A. (2016). Variability and trends in Southern Ocean eddy activity in 1/12° ocean model simulations. *Geophys. Res. Lett.*, 43(9):4517–4523.
- Saenko, O. A., Fyfe, J. C., and England, M. H. (2005). On the response of the oceanic wind-driven circulation to atmospheric CO₂ increase. *Climate dynamics*, 25(4):415–426.
- Shi, J.-R., Talley, L. D., Xie, S.-P., Peng, Q., and Liu, W. (2021). Ocean warming and accelerating Southern Ocean zonal flow. *Nat. Clim. Chg.*, 11(12):1090–1097.
- Sigmond, M., Reader, M. C., Fyfe, J. C., and Gillett, N. P. (2011). Drivers of past and future Southern Ocean change: Stratospheric ozone versus greenhouse gas impacts. *Geophys. Res. Lett.*, 38(12).
- Tamsitt, V., Talley, L. D., Mazloff, M. R., and Cerovečki, I. (2016). Zonal variations in the Southern Ocean heat budget. *J. Clim.*, 29(18):6563–6579.

- Vallis, G. K. (2017). *Atmospheric and Oceanic Fluid Dynamics*. Cambridge University Press, Cambridge, second edition.
- Watts, D. R., Tracey, K. L., Donohue, K. A., and Chereskin, T. K. (2016). Estimates of eddy heat flux crossing the Antarctic Circumpolar Current from observations in Drake Passage. *J. Phys. Oceanogr.*, 46(7):2103–2122.
- Xu, X., Chassignet, E. P., Firing, Y. L., and Donohue, K. (2020). Antarctic Circumpolar Current Transport Through Drake Passage: What Can We Learn From Comparing High-Resolution Model Results to Observations? *J. Geophys. Res. Oceans*, 125(7):e2020JC016365.

REVIEWERS' COMMENTS

Reviewer #1 (Remarks to the Author):

The authors carefully and adequately addressed all my previous concerns. I only have one or two minor suggestions.

1. It seems useful to quantify the trends by adding some numbers. Specifically, to add the trend (in Sv/year) for the acceleration between the SACCF and PF, and deceleration of the southern Flank of the Polar Front in lines 166-174. This would be a good contrast to the (lack of) trend of the net transport (Figure 3a). I assume this is essentially a partial summation of the trends in Figure 4c. It would be even better to add a figure (for example, in the supplementary) that is similar to Figure 3a but for the acceleration/deceleration parts, to highlight the trends easier than Figure 4a (but I am leaving to the author to decide if they want to add or not).

2. Given the noise or variability as shown in Figures 3a and 4a, it seems useful if the authors can comment a bit on the uncertainties of the estimated transports. The authors have discussed the error on velocities in Section 8 lines 371-376. It would be useful to have some estimates on the uncertainty of the transports, even if it is a rough estimate.

Reviewer #2 (Remarks to the Author):

The authors have addressed all my comments. I only have one minor suggestion which is rearranging the x labels for Figure 7 (b) and (c), so that they are not so close to each other to avoid confusion. Same comment for 7(e) and (f).

Response to Reviewers' Comments

NCOMMS-23-17942A

Manuel O. Gutierrez-Villanueva^a, Teresa K. Chereskin, and Janet Sprintall

^amog002@ucsd.edu

October 30, 2023

Thank you for reviewing our manuscript. Below, we have provided detailed point-by-point replies. We have used **bold and italic font for the Reviewers' comments** whereas our responses are written in normal font. In our responses, we used **text in blue** to indicate that text has been incorporated into the revised manuscript. **Text in red** in the accompanied manuscript with track changes indicates that text has been removed from the original manuscript.

Comments by Reviewer 1

The authors carefully and adequately addressed all my previous concerns. I only have one or two minor suggestions.

1: It seems useful to quantify the trends by adding some numbers. Specifically, to add the trend (in Sv/year) for the acceleration between the SACCF and PF, and deceleration of the southern Flank of the Polar Front in lines 166-174. This would be a good contrast to the (lack of) trend of the net transport (Figure 3a). I assume this is essentially a partial summation of the trends in Figure 4c. It would be even better to add a figure (for example, in the supplementary) that is similar to Figure 3a but for the acceleration/deceleration parts, to highlight the trends easier than Figure 4a (but I am leaving to the author to decide if they want to add or not).

Reply: We have added text to the end of paragraph 4 in section 4 that discusses the sum of the trends found in Fig. 4c,f,i. The added text reads:

Lines 175-180: The geostrophic and reference transport trends (Fig. 4f,i) summed across Drake Passage yield $-0.10 \text{ Sv year}^{-1}$ and $0.04 \text{ Sv year}^{-1}$, respectively, and are in close agreement with the insignificant trends calculated from the time series of net Drake Passage transport (Fig. S1). The sum of the total transport trends (Fig. 4c) yields $0.08 \text{ Sv year}^{-1}$, which is of opposite sign (but insignificant) compared to that estimated from the net transport time series (Fig. S1).

We have decided not to include the suggested figure by the reviewer in the supplementary information, since we believe the trends in Fig. 4 already show that the compensating trends coincide with the different frontal

regions. Nonetheless, we have calculated the trends in total, geostrophic, and reference transport per frontal region and included it in this response (Fig. R1). We have included the regions between the SAF and the South American continental slope (SA slope) and the region between the SACCF and the Antarctic slope (ANT slope). Trends in transport (Fig. R1) for each component show that the Polar Front compensates for the acceleration found in the region between the Subantarctic Front and Polar Front (as shown in Fig. 4); these two regions show the largest trends in Drake Passage frontal regions. The sum of the trends for each component yields no significant trend in net total transport, consistent with our results.

Fig. R1 Trends in (a) total, (b) geostrophic, and (c) reference transport per frontal region [Sv year^{-1}] for the period of October 2005 - April 2019. Filled circles are statistically significant trends ($p < 0.05$) after the modified Mann-Kendall test. Red and green circles in (a) are trends estimated using all transects and those transects with a coincident temperature/salinity (XBT/XCTD) transect, respectively. SAF, PF and SACCF are the Subantarctic Front, Polar Front, and Southern ACC Front. SA slope is the region between SAF and the South American continental slope. ANT Slope is the region between SACCF and the Antarctica continental slope. Sum is the integral of the trends per frontal regions.

2: Given the noise or variability as shown in Figures 3a and 4a, it seems useful if the authors can comment a bit on the uncertainties of the estimated transports. The authors have discussed the error on velocities in Section 8 lines 371-376. It would be useful to have some estimates on the uncertainty of the transports, even if it is a rough estimate.

Reply: We thank the reviewer for the point raised here. In addressing it, we found an inaccuracy in the Methods section where we described the velocity data processing (lines 369-371). The velocity data used for the means and trends in transport were 15-min averages (i.e., 4.5 km along-track resolution) instead of 5-min (1.5 km resolution). The oversight occurred because there are two ADCP datasets that differ only in their time resolution. We thought that all calculations were made with the 5-min datasets, but the earlier calculations of the means (Figs. 1 and S1) and trends (Figs. 2-5, and Fig. S2) employed the 15-min averages. For the trends in eddy activity (Fig. 7), the 5-min datasets were employed. Therefore, we have modified the methods section to clarify that we used the 15-min averages. The text reads:

Lines 372-383: Velocities were transformed from ship-relative to absolute ocean currents using GPS position and attitude measurements. The absolute current velocities were further averaged to 15-min resolution, i.e. ~ 4.50 km along-track resolution (assuming a ship's speed of 5 m s^{-1}). Barotropic tidal currents were removed from the absolute velocity by subtracting the tidal prediction of the TPXO7.2 tide model (Egbert et al., 1994). A slab layer was assumed for the upper 46 m where the OS38 ADCP did not sample. The single ping velocity uncertainty is 23 cm s^{-1} ; thus the uncertainty of a 300 ping average is 1.3 cm s^{-1} . Combined with the error in ship speed from GPS (0.8 cm s^{-1}) the error in absolute velocity in a 15-min average is 1.5 cm s^{-1} . Averaging over 83 min (i.e., ~ 25 km along-track assuming an average ship speed of 5 m s^{-1}) reduces the error σ_{tot} to 0.65 cm s^{-1} .

Similarly, we have modified the methods for calculating the eddy kinetic energy and momentum fluxes to state that the 5-min averages were used. The text reads:

Lines 542-546: To quantify trends in the eddy activity in Drake Passage and the eddy momentum transfer with the ACC jets, we estimated time series of depth-averaged eddy kinetic energy and eddy momentum flux from the 5-min (100 pings; 1.5 km along-track resolution) OS38 ADCP records corrected for transducer misalignment angle (see supplementary information eq. 4).

We have included a discussion about the uncertainty in the transport estimates in subsection 8 (Total, geostrophic and reference transport) estimated using the absolute error velocity σ_{tot} (Lines 382-383). The added text reads:

Lines 447-452: The error in absolute transport in the upper 760 m per 25-km along-track bin (dx) is estimated from the absolute velocity error (σ_{tot}) as $\sigma_{tot} * dx * H / \sqrt{N}$, where $H = 760$ m and N is the number of observations (transects), yielding 8×10^{-3} Sv for 237 transects for the October 2005-April 2019 period. Similarly, the standard error in absolute net Drake Passage total transport, estimated as $\sigma_{tot} * L * H / \sqrt{N}$, is 0.28 Sv.

Comments by Reviewer 2

The authors have addressed all my comments.

1: I only have one minor suggestion which is rearranging the x labels for Figure 7 (b) and (c), so that they are not so close to each other to avoid confusion. Same comment for 7(e) and (f).

Reply: We have modified Figure 7 to re-arrange the x-labels in subplots (b)-(d) and (e)-(f). The modified figure is shown below.

Fig. 7 Time series of eddy kinetic energy (EKE) and eddy momentum fluxes (EMF), mean and trends per pair of sea surface height (SSH) streamlines [m] for October 2005-April 2019. Time series of depth-averaged total (a) $\langle \text{EKE} \rangle$ and (d) $\langle \text{EMF} \rangle$ in the upper 760 m [$\text{m}^2 \text{s}^{-2}$]. Mean (b) $\overline{\langle \text{EKE} \rangle}$ and (e) $\overline{\langle \text{EMF} \rangle}$ [$\text{m}^2 \text{s}^{-2}$]. Trends in (c) $\langle \text{EKE} \rangle$ and (f) $\langle \text{EMF} \rangle$ [$\text{m}^2 \text{s}^{-2} \text{year}^{-1}$]. Red and gray lines (b)-(c) show the mean and trends estimated for all (237) available transects and only those (60) velocity transects that have a concurrent temperature/salinity transect, respectively. Filled squares (c),(f) indicate that trends are significant at 95% confidence after the modified Mann-Kendall test ($p < 0.05$). Standard errors for each line are shown in shaded areas in (b),(e). Shaded rectangles show the streamwise intervals for the Subantarctic Front (SAF), Polar Front (PF), and Southern ACC Front (SACCF) (Gutierrez-Villanueva et al., 2020).

References

- Egbert, G. D., Bennett, A. F., and Foreman, M. G. (1994). TOPEX/Poseidon tides estimated using a global inverse model. *J. Geophys. Res.*, 99(C12):24821–24852.
- Gutierrez-Villanueva, M. O., Chereskin, T. K., and Sprintall, J. (2020). Upper-ocean eddy heat flux across the Antarctic Circumpolar Current in Drake Passage from observations: time-mean and seasonal variability. *J. Phys. Oceanogr.*, 50(9):2507–2527.